# NEURAL CONTEXT FLOWS FOR META-LEARNING OF DYNAMICAL SYSTEMS

**Roussel Desmond Nzoyem**
School of Computer Science
University of Bristol
rd.nzoyemngueguin@bristol.ac.uk

**David A.W. Barton**
School of Engineering Mathematics and Technology
University of Bristol
david.barton@bristol.ac.uk

**Tom Deakin**
School of Computer Science
University of Bristol
tom.deakin@bristol.ac.uk

## ABSTRACT

Neural Ordinary Differential Equations (NODEs) often struggle to adapt to new dynamic behaviors caused by parameter changes in the underlying physical system, even when these dynamics are similar to previously observed behaviors. This problem becomes more challenging when the changing parameters are unobserved, meaning their value or influence cannot be directly measured when collecting data. To address this issue, we introduce Neural Context Flow (NCF), a robust and interpretable Meta-Learning framework that includes uncertainty estimation. NCF uses Taylor expansion to enable contextual self-modulation, allowing context vectors to influence dynamics from other domains while also modulating themselves. After establishing theoretical guarantees, we empirically test NCF and compare it to related adaptation methods. Our results show that NCF achieves state-of-the-art Out-of-Distribution performance on 5 out of 6 linear and non-linear benchmark problems. Through extensive experiments, we explore the flexible model architecture of NCF and the encoded representations within the learned context vectors. Our findings highlight the potential implications of NCF for foundational models in the physical sciences, offering a promising approach to improving the adaptability and generalization of NODEs in various scientific applications.

## 1 INTRODUCTION

A prototypical autonomous dynamical system describes the continuous change, through time $t \in \mathbb{R}^+$, of a quantity $x(t) \in \mathbb{R}^d$. Its dynamics are influenced by its parameters $c \in \mathbb{R}^{d_c}$, according to the (ordinary) differential equation

$$\frac{\mathrm{d}x}{\mathrm{d}t}(t) = f(x(t), c), \tag{1}$$

where $f : \mathbb{R}^d \times \mathbb{R}^{d_c} \to \mathbb{R}^d$ is the **vector field**. Learning a dynamical system from data is synonymous with approximating $f$, a task neural networks have been remarkably good at in recent years (Chen et al., 2018; Kidger, 2022).

Consider the challenge of reconstructing the mechanical motion of an undamped pendulum given limited data from two distinct **environments**: Earth and Mars. Disregarding the physical variations between these environments (e.g., gravitational constants, ambient temperatures, etc.), one might employ a One-For-All (OFA) approach to learn a single environment-agnostic vector field from all available data (Yin et al., 2021a). This model would struggle to adequately fit such

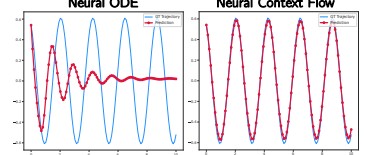

Figure 1: Predicted angle $\alpha$ for the simple pendulum $\frac{\mathrm{d}^2\alpha}{\mathrm{d}t^2} + g\sin\alpha = 0$. The OFA Neural ODE that disregards context fails to generalize its oscillation frequency $\sqrt{g}/2\pi$ to unseen environments, due in part to merged inhomogeneous training data. Our work investigates Neural Context Flows and related Meta-Learning methods to overcome this issue.

heterogeneous data and would face difficulties generalizing to data from novel environments, e.g., the Moon. Fig. 1 illustrates this issue, using gravity as the underlying physical parameter. Alternatively, learning individual vector fields for each environment with a One-Per-Env (OPE) approach would miss inter-domain commonalities, proving both time-intensive and inhibiting rapid adaptation to new environments (Yin et al., 2021a). Given these constraints, it becomes imperative to develop a methodology that can effectively *learn what to learn* from the aggregate data while simultaneously accounting for the *unique* properties of each environment.

In Scientific Machine Learning (Cuomo et al., 2022) (SciML), the problem of **generalization** has largely been tackled by injecting domain knowledge. It is commonly understood that adding a term in Eq. (1) that captures as much of the dynamics as possible leads to lower evaluation losses (Yin et al., 2021b). For such terms to be added, however, it is essential to have knowledge of the physical parameters that change, which may then either be directly estimated, or predicted by a neural network within the vector field (Rackauckas et al., 2020). We are naturally left to wonder how to efficiently learn a generalizable dynamical system when such physics are absent.

Under constantly changing experimental conditions, two major obstacles to learning the parameter dependence of a vector field can be isolated: • (P1) **Limited data** – SciML models can be data-intensive (Hey et al., 2020; Yin et al., 2021b), and limited data in each environment may not be enough to learn a vector field suitable for all environments; • (P2) **Unobserved parameters** – during the data collection and modeling processes, one might be unfamiliar with the basic physics of the system. Solving these two problems would contribute to the efficient generalization of the learned models, particularly to related but Out-of-Distribution (OoD) datasets. Fast OoD adaptation would massively reduce cost and complement recent efforts towards foundational models in the physical sciences (Subramanian et al., 2024; McCabe et al., 2023; Herde et al., 2024).

Neural Ordinary Differential Equations (Neural ODEs) (Chen et al., 2018; Weinan, 2017; Haber & Ruthotto, 2017) have emerged as a powerful backbone for learning ordinary, stochastic, and controlled differential equations (Kidger et al., 2020). Trained using Differentiable Programming techniques (Nzoyem et al., 2023; Ma et al., 2021; Rackauckas et al., 2020), they have demonstrated broad utility with outstanding results in areas like chemical engineering (Owoyele & Pal, 2022), geosciences (Shen et al., 2023), and climate modeling (Kochkov et al., 2024). In the increasingly popular SciML subfield of solving parametric PDEs (Li et al., 2020; Takamoto et al., 2023; Li et al., 2023; Subramanian et al., 2024; Koupaï et al., 2024), Neural ODEs occupy a place of choice due to their flexibility and performance (Yin et al., 2021a; Lee & Parish, 2021; Kirchmeyer et al., 2022). That said, existing methods seeking to generalize Neural ODEs to various parameter-defined environments fail to leverage information from environments other than the ones of interest. Not to mention the pervasive lack of interpretability and ways of accounting for the model's uncertainty.

This work presents **Neural Context Flows** (NCFs), a novel approach for multi-environment generalization of dynamical systems based on Neural ODEs. By leveraging the regularity of the vector field with respect to unobserved parameters, NCFs parameterize an environment-agnostic vector field and environment-specific latent context vectors to modulate the vector field. The vector field is Taylor-expanded about these context vectors, effectively allowing information to flow between environments. Our contribution is threefold:

(1) We introduce a Meta-Learning methodology for enhancing the generalizability of dynamical systems. Our approach effectively addresses problems (P1) and (P2), challenging the prevailing notion that standard Deep Learning techniques are inherently inefficient for Out-of-Distribution (OoD) adaptation (Mouli et al., 2024).

(2) We present an *interpretable* framework for Multi-Task representation learning, incorporating a straightforward method for *uncertainty estimation*. This work extends the emerging trend of explainable linearly parameterized physical systems (Blanke & Lelarge, 2024) to non-linear settings, thus broadening its applicability. For affine systems, we provide a lean proof for the identifiability of their underlying parameters.

(3) We provide a curated set of benchmark problems specifically designed for Meta-Learning of dynamical systems. This collection encompasses a diverse range of problems frequently encountered in the physical science literature, all accessible through a *unified* interface.

## 2 RELATED WORK

Learning data-driven Neural ODEs that generalize across parameters is only a recent endeavor. To the best of the authors' knowledge, all attempts to solve (P1) and (P2) have relied on Multi-Task and Meta-Learning to efficiently adapt to new parameter values, thus producing methods with varying levels of interpolation and extrapolation capabilities.

**Multi-Task Learning.** Multi-Task Learning (MTL) describes a family of techniques where a model is trained to jointly perform multiple tasks. In Scientific Machine Learning, one of the earliest methods to attack this generalization problem is LEADS (Yin et al., 2021a). In LEADS, the vector field is decomposed into shared dynamics $f_\phi$ and environment-specific $g_\psi^e$ components

$$\frac{\mathrm{d}x^e}{\mathrm{d}t} = f_\phi(x) + g_\psi^e(x), \tag{2}$$

where the superscript $e$ identifies the environment in which the dynamical system evolves, and $\{\phi, \psi\}$ are learnable neural network weights.

While LEADS excels at interpolation tasks, it performs poorly during extrapolation (Kirchmeyer et al., 2022). Furthermore, it requires retraining a new network $g_\psi^e$ each time a new environment is encountered, which can be constraining in scenarios where adaptation is frequently required. Before LEADS, other MTL approaches had been proposed outside the context of dynamical systems (Caruana, 1997; Rebuffi et al., 2017; 2018; Lee & Parish, 2021). Still, they do not address the crucial adaptation to new tasks which is our focus.

**Meta-Learning.** Another influential body of work looked at Meta-Learning: a framework in which, in addition to the MTL joint training scheme, shared representation is learned for rapid adaptation to unseen tasks with only minimum data (Wang et al., 2021). The seminal MAML (Finn et al., 2017) popularized Gradient-Based Meta-Learning (GBML) by nesting an inner gradient loop in the typical training process. Since then, several variants aimed at avoiding over-fitting and reducing cost have been proposed, e.g. ANIL (Raghu et al., 2019) and CAVIA (Zintgraf et al., 2019). The latter is a contextual learning approach (Garnelo et al., 2018) that partitions learnable parameters into some that are optimized on each environment, and others shared across environments, i.e., *meta-trained*.

DyAd (Wang et al., 2022) is one of the earliest Meta-Learning approaches to target generalizable dynamical systems. It learns to represent time-invariant features of a trajectory by an *encoder* network, followed by a *forecaster* network to learn shared dynamics across the different environments. DyAd only performs well under weak supervision, that is when the underlying (observed) parameters are made known to the loss function via penalization.

Arguably the most successful Meta-Learning method for physical systems is CoDA (Kirchmeyer et al., 2022), which assumes that the underlying system is described by parametrized differential equations whose form is shared by all environments. However, these equations differ by the values of the vector field's weights, which are produced by a (linear) *hypernetwork*. For the environment indexed by $e$, these weights are computed by[1]

$$\theta^e = \theta^c + W\xi^e, \tag{3}$$

where $\theta^c$ and $W$ are shared across environments, and $\xi^e \in \mathbb{R}^{d_\xi}$ is an environment-specific latent context vector (or simply **context**). While it achieves state-of-the-art performance on many physical systems, the main limitation of CoDA is its hypernetwork approach, which might hinder parallelism and memory scaling to large root or target networks. In practice, methods based on hypernetworks require more computational resources to backpropagate and train, and exhibit a more complex optimization landscape (Chauhan et al., 2023).

**Taylor Expansion.** This local approximation strategy finds numerous applications in SciML, notably helping establish single-trajectory geometrical requirements for linear and affine system identification (Duan et al., 2020). Recently, Blanke & Lelarge (2024) modeled the variability of

---

[1]The general CoDA formulation encompasses the GMBL adaptation rule (Kirchmeyer et al., 2022). Furthermore, MTL models can be identified to Eq. (3) with $\theta^c = \mathbf{0}$.

linearly parameterized dynamical systems with an affine function of low-dimensional environment-specific context vectors. They empirically showed that this improved interpretability, generalization abilities, and computation speed. In our work, we similarly explain *non-linear* systems, thus generalizing existing work with higher-order Taylor expansion. Additionally, we extract benefits such as massive parallelizability and uncertainty estimation.

## 3    NEURAL CONTEXT FLOW

A training dataset $\mathcal{D}_{\mathrm{tr}} := \left\{ x_i^e(\cdot) \right\}_{i \in [\![ 1,S ]\!]}^{e \in [\![ 1,m ]\!]}$ is defined as a set of trajectories collected from $m$ related environments, with $S$ trajectories per environment, each of length $N \in \mathbb{N}^*$ over a time horizon $T > 0$. Given $\mathcal{D}_{\mathrm{tr}}$, we aim to find the neural network weights $\theta$ that parameterize a vector field $f_\theta$, along with several context vectors $\{\xi^e\}_{e \in [\![ 1,m ]\!]}$ that modulate its behavior such that

$$\frac{\mathrm{d}x_i^e}{\mathrm{d}t}(t) = f_\theta(x_i^e(t), \xi^e), \qquad \forall t \in [0,T], \quad \forall i \in [\![ 1,S ]\!], \quad \forall e \in [\![ 1,m ]\!]. \tag{4}$$

We learn a single vector field for all environments in our training set $\mathcal{D}_{\mathrm{tr}}$. The same vector field will be reused, unchanged, for future testing and adaptation to environments in a similarly-defined $\mathcal{D}_{\mathrm{ad}}$.

The vector field $f_\theta$ is assumed to not only be *continuous*, but also *smooth* in its second argument $\xi$. Exploiting this constraint, we "collect" information from other environments by Taylor-expanding $f_\theta$ around any other $\{\xi^j\}_{j \in \mathrm{P}}$, where $\mathrm{P} \subseteq [\![ 1,m ]\!]$ is a **context pool**[2] containing $p := |\mathrm{P}| \geq 1$ environment indices. This gives rise, for fixed $e$ and $i$, to $p$ Neural ODEs

$$\begin{cases} \dfrac{\mathrm{d}x_i^{e,j}}{\mathrm{d}t}(t) = T_{f_\theta}^k(x_i^{e,j}(t), \xi^e, \xi^j), \\ x_i^{e,j}(0) = x_i^e(0), \end{cases} \qquad \forall j \in \mathrm{P}, \tag{5}$$

where $x_i^{e,j}(\cdot) \in \mathbb{R}^d$, and $\xi^e, \xi^j \in \mathbb{R}^{d_\xi}$. $T_{f_\theta}^k(\cdot, \xi^e, \xi^j)$ denotes the $k$-th order Taylor expansion of $f_\theta$ at $\xi^e$ around $\xi^j$. In particular, $T_{f_\theta}^1$ can be written as

$$T_{f_\theta}^1(x_i^{e,j}, \xi^e, \xi^j) = f_\theta(x_i^{e,j}, \xi^j) + \nabla_\xi f(x_i^{e,j}, \xi^j)(\xi^e - \xi^j) + o(\|\xi^e - \xi^j\|), \tag{6}$$

where $o(\cdot)$ captures negligible residuals. Eq. (6) directly consists of a Jacobian-Vector Product (JVP), making its implementation memory-efficient. Since higher-order Taylor expansions of vector-valued functions do not readily display the same property, we provide the following proposition to facilitate the second-order approximation.

**Proposition 1** (Second-order Taylor expansion with JVPs). *Assume $f : \mathbb{R}^d \times \mathbb{R}^{d_\xi} \to \mathbb{R}^d$ is $\mathcal{C}^2$ wrt its second argument. Let $x \in \mathbb{R}^d, \xi \in \mathbb{R}^{d_\xi}$, and define $g : \bar{\xi} \mapsto \nabla_\xi f(x, \bar{\xi})(\xi - \bar{\xi})$. The second-order Taylor expansion of $f$ around any $\tilde{\xi} \in \mathbb{R}^\xi$ is then expressed as*

$$f(x, \xi) = f(x, \tilde{\xi}) + \frac{3}{2}g(\tilde{\xi}) + \frac{1}{2}\nabla g(\tilde{\xi})(\xi - \tilde{\xi}) + o(\|\xi - \tilde{\xi}\|^2). \tag{7}$$

*Proof.* We refer the reader to Appendix A. $\qquad\square$

By setting $\xi^e := \xi$, and $\xi^j := \tilde{\xi}$, Proposition 1 yields an expression for $T_{f_\theta}^2(\cdot, \xi^e, \xi^j)$ in terms of JVPs, an implementation of which we detail in Appendix E. During training (as described below) trajectories from all $p$ Neural ODEs are used within the loss function.

This new framework is called **Neural Context Flow** (NCF) as is the resulting model. The "flow" term refers to the capability of the context from one environment, i.e., $j$, to influence predictions in another environment, i.e., $e$, by means of Taylor expansion. It allows the various contexts to not only modulate the behavior of the vector field, but to equally modulate theirs, since they are forced to remain close for the Taylor approximations to be accurate. This happens while the same contexts are pushed apart by the diversity in the data. This self-modulation process and the beneficial friction

---

[2]We note that P might include $e$ itself, and is reconstituted at each evaluation of Eq. (9). Its size $p$ is constant, each element indexing a *distinct* environment for computational efficiency (see Appendix A.6).

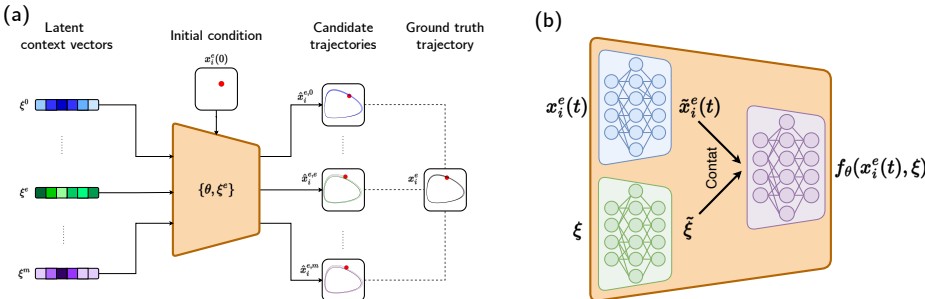

Figure 2: **(a)** Illustration of the Neural Context Flow (NCF). Given an initial condition $x_i^e(0)$ for a training trajectory $i$ in the environment $e$, NCF predicts in parallel, several candidate trajectories $\{\hat{x}_i^{e,j}\}_{j\in\text{P}}$ that are all compared to the ground truth $x_i^e$, upon which $\{\theta, \xi^e\}$ is updated. **(b)** Depiction of the 3-networks architecture for $f_\theta$, where the state vector and an arbitrary context vector are projected into the same representational space before they can interact inside the main network.

it creates is notably absent from other contextual Meta-Learning approaches like CAVIA (Zintgraf et al., 2019) and CoDA (Kirchmeyer et al., 2022).

We depict the NCF framework in Fig. 2, along with a **3-networks** architecture that lifts the contexts and the state vectors into the same representational space before they can interact. Concretely, $x_i^e(t)$ and the context $\xi$ are first processed independently into $\tilde{x}_i^e(t)$ (by a state network in blue) and $\tilde{\xi}$ (by a context network in green) respectively, before they are concatenated and fed to a main network (in purple) to produce $f_\theta(x_i^e(t), \xi)$. This explicitly allows the model to account for the potentially nonlinear relationship between the context and the state vector at each evaluation of the vector field.

### 3.1 META-TRAINING

Starting from the same initial state $\hat{x}_i^{e,j}(0) := x_i^e(0)$, the Neural ODEs in Eq. (5) are integrated using a differentiable numerical solver (Kidger, 2022; Nzoyem et al., 2023; Poli et al.; Chen, 2018):

$$\hat{x}_i^{e,j}(t) = \hat{x}_i^{e,j}(0) + \int_0^t T_{f_\theta}^k(\hat{x}_i^{e,j}(\tau), \xi^e, \xi^j)\,\mathrm{d}\tau, \quad \forall j \in \text{P}. \tag{8}$$

The resulting **candidate** trajectories are evaluated at specific time steps $\{t_n\}_{n\in[\![1,N]\!]}$ such that $t_1 = 0$ and $t_N = T$. We feed these to a supervised (inner) loss function

$$\ell(\theta, \xi^e, \xi^j, \hat{x}_i^{e,j}, x_i^e) := \frac{1}{N \times d} \sum_{n=1}^N \|\hat{x}_i^{e,j}(t_n) - x_i^e(t_n)\|_2^2 + \frac{\lambda_1}{d_\xi}\|\xi^e\|_1 + \frac{\lambda_2}{d_\theta}\|\theta\|_2^2, \tag{9}$$

where $d, d_\xi$, and $d_\theta$ are the dimensions of the state space, context vectors, and flattened network weights, respectively. $\|\cdot\|_1$ and $\|\cdot\|_2$ denote the regularizing $L^1$ and $L^2$ norms, with $\lambda_1$ and $\lambda_2$ their penalty coefficients, respectively. To ensure its independence from environment count, context pool size, and trajectory count per environment, the overall MSE loss function $\mathcal{L}$ is expressed as in Eq. (10); after which it is minimized wrt both weights $\theta$ and contexts $\xi^{1:m} := \{\xi^e\}_{e\in[\![1,m]\!]}$ via gradient descent, alternating between updates:

$$\mathcal{L}(\theta, \xi^{1:m}, \mathcal{D}_{\text{tr}}) := \frac{1}{m \times S \times p} \sum_{e=1}^m \sum_{i=1}^S \sum_{j=1}^p \ell(\theta, \xi^e, \xi^j, \hat{x}_i^{e,j}, x_i^e). \tag{10}$$

The most effective way to train NCFs is via **proximal** alternating minimization as described in Algorithm 1. Although more computationally demanding compared to ordinary alternating minimization (see Algorithm 3), it is adept at dealing with non-smooth loss terms like the $L^1$ norm in Eq. (9) (Parikh et al., 2014). Not to mention that ordinary alternating minimization (Algorithm 3) can easily lead to sub-optimal convergence (Attouch et al., 2010; Li et al., 2019), while its proximal counterpart converges, with random initialization, almost surely to second-order stationary points provided mild

assumptions are satisfied (see Theorem 1). We provide a short discussion on those assumptions in Appendix A.3.

The above comparison demands the definition of two variants of Neural Context Flows. **NCF-**$t_2$ uses *second*-order Taylor expansion in equation 5, and is trained via *proximal* alternating minimization (Algorithm 1). **NCF-**$t_1$ on the other hand, is implemented using a *first*-order Taylor expansion, and trained using *ordinary* alternating minimization (Algorithm 3). Although less expressive than NCF-$t_2$, NCF-$t_1$ is faster and serves as a powerful baseline in our experiments.

**Theorem 1** (Convergence to second-order stationary points)**.** *Assume that $\mathcal{L}(\cdot, \cdot, \mathcal{D}_{tr})$ satisfies the Kurdyka-Lojasiewicz (KL) property, is $L$ bi-smooth, and $\nabla \mathcal{L}(\cdot, \cdot, \mathcal{D}_{tr})$ is Lipschitz continuous on any bounded subset of domain $\mathbb{R}^{d_\theta} \times \mathbb{R}^{d_\xi \times m}$. Under those assumptions, let $(\theta_0, \xi_0^{1:m})$ be a random initialization and $(\theta_q, \xi_q^{1:m})$ be the sequence generated by Algorithm 1. If the sequence $(\theta_q, \xi_q^{1:m})$ is bounded, then it converges to a second-order stationary point of $\mathcal{L}(\cdot, \cdot, \mathcal{D}_{tr})$ almost surely.*

---

**Algorithm 1** Proximal Alternating Minimization

1: **Input:** $\mathcal{D}_{tr} := \{\mathcal{D}^e\}_{e \in [\![1,m]\!]}$
2: $\theta_0 \in \mathbb{R}^{d_\theta}$ randomly initialized
3: $\xi_0^{1:m} = \bigcup\limits_{e=1}^{m} \xi^e$, where $\xi^e = \mathbf{0} \in \mathbb{R}^{d_\xi}$
4: $q_{max} \in \mathbb{N}^*; \beta \geq L \in \mathbb{R}^+; \eta_\theta, \eta_\xi > 0$
5: **for** $q \leftarrow 1, q_{max}$ **do**
6: $\quad \mathcal{G}(\theta) := \mathcal{L}(\theta, \xi_{q-1}^{1:m}, \mathcal{D}_{tr}) + \frac{\beta}{2}\|\theta - \theta_{q-1}\|_2^2$
7: $\quad \theta_q = \theta_{q-1}$
8: $\quad$ **repeat**
9: $\quad\quad \theta_q \leftarrow \theta_q - \eta_\theta \nabla \mathcal{G}(\theta_q)$
10: $\quad$ **until** $\theta_q$ converges
11: $\quad \mathcal{H}(\xi^{1:m}) := \mathcal{L}(\theta_q, \xi^{1:m}, \mathcal{D}_{tr})$
$\qquad\qquad\qquad + \frac{\beta}{2}\|\xi^{1:m} - \xi_{q-1}^{1:m}\|_2^2$
12: $\quad \xi_q^{1:m} = \xi_{q-1}^{1:m}$
13: $\quad$ **repeat**
14: $\quad\quad \xi_q^{1:m} \leftarrow \xi_q^{1:m} - \eta_\xi \nabla \mathcal{H}(\xi_q^{1:m})$
15: $\quad$ **until** $\xi_q^{1:m}$ converges
16: **end for**

---

**Algorithm 2** Sequential Adaptation of NCF

1: **Input:** $\mathcal{D}_{ad} := \{\mathcal{D}^{e'}\}_{e' \in [\![a,b]\!]}$
2: $\theta \in \mathbb{R}^{d_\theta}$ learned
3: $\xi^{e'} = \mathbf{0} \in \mathbb{R}^{d_\xi}, \forall e' \in [\![a,b]\!]$
4: $\eta > 0$
5: **for** $e' \leftarrow a, b$ **do**
6: $\quad \mathcal{L}(\xi^{e'}, \mathcal{D}^{e'}) :=$
$\qquad\qquad \frac{1}{S'} \sum\limits_{i=1}^{S'} \ell(\theta, \xi^{e'}, \xi^{e'}, \hat{x}_i^{e',e'}, x_i^{e'})$
7: $\quad$ **repeat**
8: $\quad\quad \xi^{e'} \leftarrow \xi^{e'} - \eta \nabla \mathcal{L}(\xi^{e'}, \mathcal{D}^{e'})$
9: $\quad$ **until** $\xi^{e'}$ converges
10: **end for**

*Proof.* The proof of Theorem 1 is straightforward by adapting Assumptions 1 and 4, then Theorem 2 of (Li et al., 2019). $\qquad\square$

---

## 3.2 Adaptation (or Meta-Testing)

Few-shot adaptation to a new environment $e' \in [\![a, b]\!]$ in or out of the meta-training distribution requires relatively less data, i.e., its trajectory count $S' \ll S$. Here, the network weights are frozen, and the goal is to find a context $\xi^{e'}$ such that

$$\frac{\mathrm{d}x_i^{e'}}{\mathrm{d}t}(t) = f_\theta(x_i^{e'}(t), \xi^{e'}), \qquad \forall i \in [\![1, S']\!]. \tag{11}$$

Our adaptation rule, as outlined in Algorithm 2 is extremely fast, converging in seconds for trainings that took hours. In scenarios where we want to adapt to more than one environment, we outline a bulk version in Appendix A.5 (see Algorithm 4). Although the bulk adaptation algorithm is parallelizable and framed in the same way as during meta-training, it does not allow for flow of contextual information, since this causes significant accuracy degradation. Most importantly, disabling the Taylor expansion at this stage limits the size of the context pool to $p = 1$ and significantly improves memory efficiency, an important resource when adapting to a large number of environments.

## 3.3 InD and OoD Testing

We distinguish two forms of testing. For **In-Domain** (InD) testing, the environments are the same as the ones used during training. In InD testing data, the underlying parameters of the dynamical system we aim to reconstruct are unchanged, and so the meta-learned context vectors are reused. **Out-of-Distribution** (OoD) testing considers environments encountered during adaptation. The data

is from the same dynamical system, but defined by different parameter values (either interpolated or extrapolated). In all forms of testing, only the main predicted trajectory corresponding to $j = e$ or $j = e'$ is used in the MSE and MAPE metrics computation (see Appendix B.4 for definitions), thus returning to a standard Neural ODE. Other candidate trajectories are aggregated to ascertain the model's uncertainty. The initial conditions that define the trajectories are always unseen, although their distribution never changes from meta-training to meta-testing.

## 4 MAIN RESULTS

In this section, we evaluate the effectiveness of our proposed framework by investigating two main questions. (*i*) How good are NCFs at resolving (P1) and (P2) for interpolation tasks (Section 4.2)? (*ii*) How does our framework compare to SoTA Meta-Learning baselines (Section 4.3)? Further questions regarding interpretability, scalability, and uncertainty estimation are formulated and addressed in Appendices A.2, A.7 and B.2 respectively.

### 4.1 EXPERIMENTAL SETTING

The NCF framework is evaluated on five seminal benchmarks. The Simple Pendulum (**SP**) models the periodic motion of a point mass suspended from a fixed point. The Lotka-Volterra (**LV**) system models the dynamics of an ecosystem in which two species interact. Additional ODEs include a simple model for yeast glycolysis: Glycolytic-Oscillator (**GO**) (Daniels & Nemenman, 2015), and the more advanced Sel'kov Model (**SM**) (Strogatz, 2018; Sel'Kov, 1968). Like GO, SM is non-linearly parameterized, but additionally exhibits starkly different behaviors when its key parameter is varied (see Fig. 4(a)). Finally, we consider three PDEs, all with periodic boundary conditions, and cast as ODEs via the method of lines: the non-linear oscillatory Brusselator (**BT**) model for autocatalytic chemical reactions (Prigogine & Lefever, 1968), the Gray-Scott (**GS**) system also for reaction-diffusion in chemical settings (Pearson, 1993), and Navier-Stokes (**NS**) for incompressible fluid flow (Stokes et al., 1851).

For all problems, the parameters and initial states are sampled from distributions representative of real-world problems observed in the scientific community, and the trajectories are generated using a time-adaptive 4th-order Runge-Kutta solver (Virtanen et al., 2020). For LV, GO, GS, and NS, we reproduce the original guidelines set in (Kirchmeyer et al., 2022), while exposing the data for ODE and PDE problems alike via a common interface. Such use of synthetic data is a common practice in this emerging field of generalizable dynamical systems, where the search for unifying benchmarks remains an open problem (Massaroli et al., 2020). This need for shared datasets and APIs has motivated the `Gen-Dynamics` open-source initiative, our third and final contribution with this paper. Further details, along with the data generation process, are given in Appendix B.

We now highlight a few key practical considerations shared across experiments. We use the 3-networks architecture depicted in Fig. 2b to suitably process the state and context variables. The dimension of the context vector, the context pool's size and filling strategy, and the numerical integration scheme vary across problems. For instance, we set $d_\xi = 1024$ for LV, $d_\xi = 202$ for NS, and $d_\xi = 256$ for all other problems; while $p = 2$ for LV and SM, $p = 4$ for LV and GO, and $p = 3$ for all PDE problems. Other hyperparameters are carefully discussed in Appendix D.

### 4.2 INTERPOLATION RESULTS

This experiment explores the SP problem discussed in Fig. 1. During meta-training, we use 25 environments with the gravity $g$ regularly spaced in $[2, 24]$. Each of these environments contains only 4 trajectories with the initial conditions $x_i^e(0) \sim \left( \mathcal{U}(-\frac{\pi}{3}, \frac{\pi}{3}), \mathcal{U}(-1, 1) \right)^T$. During adaptation, we interpolate to 2 new environments with $g \in \{10.25, 14.75\}$, each with 1 trajectory. For both training and adaptation testing scenarios, we generate 32 separate trajectories.

Table 1: Training and adaptation testing MSEs ($\downarrow$) with OFA, OPE, and NCF-$t_1$ on the SP problem.

| | TRAIN ($\times 10^{-1}$) | ADAPT ($\times 10^{-3}$) |
|---|---|---|
| OFA | $9.49 \pm 0.04$ | $115000 \pm 3200$ |
| OPE | $0.18 \pm 0.02$ | $459.0 \pm 345.0$ |
| NCF | $0.10 \pm 0.03$ | $0.0356 \pm 0.001$ |

Table 2: In-Domain (InD) and adaptation (OoD) test MSEs ($\downarrow$) for the LV, GO, SM, BT, GS and NS problems. The best is reported in **bold**. The best of the two NCF variants is shaded in grey .

| | **LV** ($\times 10^{-5}$) | | | **GO** ($\times 10^{-4}$) | | |
|---|---|---|---|---|---|---|
| | #PARAMS | IND | OOD | #PARAMS | IND | OOD |
| CAVIA | 305246 | 91.0±63.6 | 120.1±28.3 | 130711 | 64.0±14.1 | 463.4±84.9 |
| CODA | 305793 | **1.40±0.13** | 2.19±0.78 | 135390 | 5.06±0.81 | 4.22±4.21 |
| NCF-$t_1$ | 308240 | 6.73±0.87 | 7.92±1.04 | 131149 | 40.3±9.1 | 19.4±1.24 |
| NCF-$t_2$ | 308240 | 1.68±0.32 | **1.99±0.31** | 131149 | **3.33±0.14** | **2.83±0.23** |

| | **SM** ($\times 10^{-3}$) | | | **BT** ($\times 10^{-1}$) | | |
|---|---|---|---|---|---|---|
| | #PARAMS | IND | OOD | #PARAMS | IND | OOD |
| CAVIA | 50486 | 979.1±141.2 | 859.1±70.7 | 116665 | 21.93±1.8 | 22.6±7.22 |
| CODA | 50547 | 156.0±40.52 | 8.28±0.29 | 119679 | 25.40±9.5 | 19.47±11.6 |
| NCF-$t_1$ | 50000 | 680.6±320.1 | 677.2±18.7 | 117502 | 21.53±8.9 | 20.89±12.0 |
| NCF-$t_2$ | 50000 | **6.42±0.41** | **2.03±0.12** | 117502 | **3.46±0.09** | **3.77±0.15** |

| | **GS** ($\times 10^{-3}$) | | | **NS** ($\times 10^{-3}$) | | |
|---|---|---|---|---|---|---|
| | #PARAMS | IND | OOD | #PARAMS | IND | OOD |
| CAVIA | 618245 | 69.9±21.2 | 68.0±4.2 | 310959 | 128.1±29.9 | 126.4±20.7 |
| CODA | 619169 | **1.23±0.14** | **0.75±0.65** | 309241 | 7.69±1.14 | 7.08±0.07 |
| NCF-$t_1$ | 610942 | 7.64±0.70 | 5.57±0.21 | 310955 | 2.98±0.09 | 2.83±0.06 |
| NCF-$t_2$ | 610942 | 6.15±0.24 | 3.40±0.51 | 310955 | **2.92±0.08** | **2.79±0.09** |

Table 1 emphasize the adaptation-time merits of Meta-Learning approaches like NCFs in lieu of baselines where one context-agnostic vector field is trained for all environments indiscriminately (One-For-All or OFA), or for each environment independently (One-Per-Env or OPE). Additional results and further analysis of these differences is done in Appendix B.3.

## 4.3 EXTRAPOLATION RESULTS

A Meta-Learning algorithm is only as good as its ability to extrapolated to unseen environments. In this section we tackle several one-shot generalization problems, i.e. $S' = 1$. We consider the LV, GO, SM, GS, BT, and NS problems, which all involve adaptation environments outside their meta-training distributions. We compare both variants of NCF to two baselines: CAVIA (Zintgraf et al., 2019) which is conceptually the closest GBML method to ours, and CoDA-$\ell_1$ (Kirchmeyer et al., 2022). Their hyperparameters, laid out in Appendix D, are tuned for a balance of computational efficacy and performance, all the while respecting each baseline's key assumptions.

At similar parameter counts[3], Table 2 shows that CAVIA is the least effective for learning all six physical systems, with a tendency to overfit on the few-shots it receives during both meta-training and meta-testing (Mishra et al., 2017). NCF-$t_2$ achieves SoTA OoD results on 5 out of 6 problems. While CoDA retains its superiority on GS, we find that it struggles on non-linear problems, most notably on the SM problem whose trajectories go through 3 distinct attractors in a Hopf bifurcation.

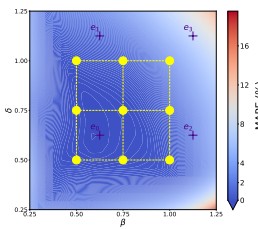

This shows that CoDA's ability to automatically select low-rank adaptation subspaces via a linear hypernetwork decoder is limited, particularly for highly nonlinear systems. We note that NCF-$t_1$ equally fails to accurately resolve SM and other non-linear problems, thus illustrating the value of second-order Taylor expansion. This suggests that higher-order Taylor-based regularization prevents NCF-$t_2$ from learning spurious associations, which is a problem commonly associated with poor OoD generalization (Mouli et al., 2024).

Figure 3: Grid-wise adaptation on the LV problem showing low MAPEs ($\downarrow$).

**Grid-wide adaptation on LV.** We report in Fig. 3 how well NCF performs on 625 adaptation environments obtained by varying its parameters

---

[3]This accounts for the total number of learnable parameters in the framework excluding the context vectors, whose size might vary with the method as explained in Appendix D.

$\beta$ and $\delta$ on a 25×25 uniform grid. It shows a consistently low MAPE below 2%, except in the bottom right corner where the MAPE rises to roughly 15%, still a remarkably low value for this problem. We highlight the 9 meta-training environments in yellow, and the 4 environments used for OoD adaptation for Table 2 in indigo. Remarkably, our training environment's convex-hull where adaptation is particularly low is much larger compared to CoDA's (Kirchmeyer et al., 2022, Figure 3).

**Nonlinear adaptation on SM.** We investigate how the models perform across the SM attractors depicted in Fig. 4(a): training environments $e_1$ and $e_2$ fall into a limit cycle (L1), $e_3$ and $e_4$ collapse to a stable equilibrium (E), while $e_5$ and $e_6$ fall into another limit cycle (L2). Fig. 4(c) presents the adaptation MSE for each environment in the OoD testing dataset. While CoDA equally fails to capture all 3 attractors, we observe that CAVIA and NCF-$t_2$ tend to favor E. But unlike CAVIA, NCF-$t_2$ succeeds in capturing limit cycles as well, as evidenced by the low MSE on all environments.

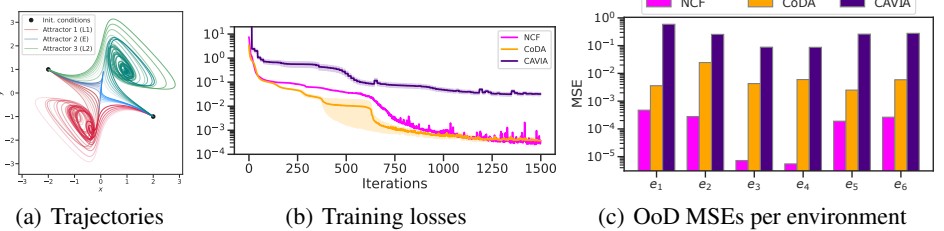

| (a) Trajectories | (b) Training losses | (c) OoD MSEs per environment |

Figure 4: (a) Sample trajectories from the SM problem illustrating the Hopf bifurcation; (b) Losses during meta-training; (c) Subsequent adaptation MSE per method per environment.

## 5 DISCUSSION

### 5.1 BENEFITS OF NCFs

Neural Context Flows provide a powerful and flexible framework for learning differential equations. They can handle irregularly-sampled sequences like time series, and can easily be extended to general regression tasks (Finn et al., 2017). Some of their other desirable properties are highlighted below.

**Massively parallelizable.** Eq. (10) indicates that NCFs are massively parallelizable along 3 directions. Indeed, evaluations of $\mathcal{L}$ can be vectorized across $m \times p$ environments, and across all $S$ trajectories. This leads to better use of computational resources for meta-training. We provide details on such vectorized NCF implementation in Appendix E, along with a thorough discussion on its scalability in Appendix A.7.

**Interpretable.** Understanding how a model adapts to new physical settings is invaluable in many scientific scenarios. Moving information from one environment to another via context-affine transformations provides a powerful framework for explaining Multi-Task Learning (Blanke & Lelarge, 2024). With contextual self-modulation via Taylor expansion, we generalize this framework while maintaining computational efficiency. In Fig. 5 for instance, we showcase system identification with NCF-$t_1$, where the underlying physical parameters $\beta$ and $\delta$ of the LV problem are recovered up to a linear transform.

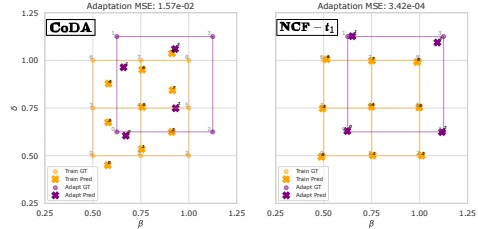

Figure 5: Interpretability with NCF.

The corresponding experiment is detailed in Appendix A.2 which further illustrates NCF's robustness to noise during system identification.

**Proposition 2** (Identifiability of affine systems). *Assume $d_\xi \geq d_c$, that $P$ is full-rank, and that $f_{\text{true}}$ is differentiable in its second argument. In the limit of zero training loss in Eq. (10), $f_\theta$ trained with first-order Taylor expansion and the Random-All pool-filling strategy[4] is affine on an open region of $\mathbb{R}^{d_\xi}$. Furthermore, there exists $Q \in \mathbb{R}^{d_c \times d_\xi}$ and $q \in \mathbb{R}^{d_c}$ such that for any meta-trained $\xi \in \{\xi^e\}_{e=1}^m$ and its corresponding underlying parameter $c \in \{c^e\}_{e=1}^m$, we have $c = Q\xi + q$.*

---

[4]A discussion on which neighboring contexts should go into the context pool P is provided in Appendix A.6.

To theoretically demonstrate the capacity to identify physical systems, we consider a learnable vector field $f_{\text{true}} : \mathbb{R}^d \times \mathbb{R}^{d_c} \to \mathbb{R}^d$ is that is affine i.e. $\exists P \in \mathbb{R}^{d \times d_c}$ and $p \in \mathbb{R}^d$ such that $f_{\text{true}}(\cdot, c) = Pc + p$. Under these assumptions, the predictor $f_\theta : \mathbb{R}^d \times \mathbb{R}^{d_\xi} \to \mathbb{R}^d$ satisfies Proposition 2 above, inspired by Proposition 1 of (Blanke & Lelarge, 2024). A detailed proof is provided in Appendix A.2.

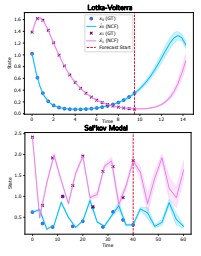

Figure 6: Uncertainty estimation with NCF.

**Extendable.** The relative simplicity of our formulation makes NCFs easily adaptable to other works involving Neural ODE models. For instance, it is straightforward to augment the vector field as in APHYNITY (Yin et al., 2021b), augment the state space as in ANODE (Dupont et al., 2019), or control the vector field as in (Massaroli et al., 2020).

**Provides uncertainty.** Uncertainty requirements are increasingly critical in Meta-Learned dynamical systems (Liu et al.). NCFs provide, by their very definition, a measure of the uncertainty in the model. During testing, all candidate trajectories stemming from available contexts (meta-trained, adapted, or both) can be used to ascertain when or where the model is most uncertain in its predictions. In Fig. 6 for instance, we visualize the means and standard deviations (scaled 10 folds for visual exposition) across 9 and 21 candidate forecasts for the Lotka-Volterra and Sel'kov Model problems, respectively. Additional results with quantitative metrics are provided in Appendix B.2.

## 5.2 LIMITATIONS

**Further theory.** While massively parallelizable, extendable, interpretable, and applicable to countless areas well outside the learning of dynamics models, Neural Context Flows still need additional theoretical analysis to explain their effectiveness and uncover failure modes. While the provided Propositions 1 and 2 partly compensates for this shortage, we believe that this creates an interesting avenue, along with other limitations outlined below, for future work.

**Regularity assumptions.** Another limitation faced by Neural Context Flows lies within the assumptions they make. While differentiability of the vector field $f$ wrt $\xi$ is encountered with the majority of dynamical systems in science and engineering, some are bound to be fundamentally discontinuous. NCFs break down in such scenarios and would benefit from the vast body of research in numerical continuation (Allgower & Georg, 2012).

**Additional hyperparameters.** Finally, NCF introduces several new hyperparameters such as the context size $d_\xi$, the context pool size $p$, the pool-filling strategy, the proximity coefficient $\beta$ in Algorithm 1, and many more. Although we offer insights into their roles in the Appendix C, we acknowledge that tuning them complicates the training process.

## 5.3 CONCLUSION AND FUTURE WORK

This paper introduces Neural Context Flows (NCFs), an innovative framework that enhances model generalization across diverse environments by exploiting the differentiability of the predictor in its latent context vector. The novel application of Taylor expansion in NCFs facilitates vector field modulation for improved adaptation, enhances interpretability, and provides valuable uncertainty estimates for deeper model understanding. Our comprehensive experiments demonstrate the robustness and scalability of the NCF approach, particularly with respect to its most demanding hyperparameters. Future research will explore the limits of NCFs and their adaptation to even more complex scenarios. This work represents a promising step toward developing foundational models that generalize across scientific domains, offering a fresh and versatile approach to conditioning machine learning models.

## ETHICS STATEMENT

While the benefits of NCFs are evidenced in Section 5.1, its negative impacts should not be neglected. For instance, malicious deployment of such adaptable models in scenarios they were not designed for could lead to serious adverse outcomes. With that in mind, our code, data, and models are openly available at `https://github.com/ddrous/ncflow`.

ACKNOWLEDGMENTS

This work was supported by UK Research and Innovation grant EP/S022937/1: Interactive Artificial Intelligence, EPSRC program grant EP/R006768/1: Digital twins for improved dynamic design, EPSRC grant EP/X039137/1: The GW4 Isambard Tier-2 service for advanced computer architectures, and EPSRC grant EP/T022205/1: JADE: Joint Academic Data science Endeavour-2. We extend our sincere gratitude to the anonymous reviewers whose insightful feedback and rigorous commentary substantially enhanced the quality and clarity of this manuscript. We also acknowledge Amarpal Sahota for valuable discussions on uncertainty quantification methodologies.

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

# SUPPLEMENTARY MATERIAL FOR NEURAL CONTEXT FLOWS FOR META-LEARNING OF DYNAMICAL SYSTEMS

# A  ALGORITHMS & PROOFS

## A.1  SECOND-ORDER TAYLOR EXPANSION WITH JVPS

For ease of demonstration, we propose an equivalent formulation of Proposition 1 that disregards the first input of $f$ not necessary for its proof. Although we will consistently write terms like $f(x)$, we emphasize that $x$ is meant to stand for the context, not the state variable.

**Proposition 1** (Second-order Taylor expansion with JVPs). *Assume $f : \mathbb{R}^{d_\xi} \to \mathbb{R}^d$ is $\mathcal{C}^2$. Let $x \in \mathbb{R}^{d_\xi}$, and define $g : y \mapsto \nabla f(y)(x - y)$. The second-order Taylor expansion of $f$ around any $x_0 \in \mathbb{R}^{d_\xi}$ is then expressed as*

$$f(x) = f(x_0) + \frac{3}{2}g(x_0) + \frac{1}{2}\nabla g(x_0)(x - x_0) + o(\|x - x_0\|^2).$$

*Proof.* Let $x, x_0 \in \mathbb{R}^{d_\xi}$. The second-order Taylor-expansion of $f$ includes its Hessian that we view as a 3-dimensional tensor, and we contract along its last axis such that

$$f(x) = f(x_0) + \nabla f(x_0)(x - x_0) + \frac{1}{2}\left[\nabla^2 f(x_0)(x - x_0)\right](x - x_0) + o(\|x - x_0\|^2). \quad (12)$$

Next we define $g : y \mapsto \nabla f(y)(x - y)$, and we consider a small perturbation $h \in \mathbb{R}^{d_\xi}$ to write

$$\begin{aligned}
g(y + h) &= \nabla f(y + h)(x - y - h) \\
&= \left[\nabla f(y) + \nabla^2 f(y)(h) + o(\|h\|)\right](x - y - h) \quad \text{☞ Taylor expansion of } \nabla f \\
&= \nabla f(y)(x - y) - \nabla f(y)h + \left[\nabla^2 f(y)h\right](x - y) + o(\|h\|) + \cancel{o(\|h\|^2)}
\end{aligned}$$

The Hessian is by definition symmetric along its last two axes, which allows us to rewrite the third term as $\left[\nabla^2 f(y)(x - y)\right]h$. We then have

$$g(y + h) = g(y) + \left[\nabla^2 f(y)(x - y) - \nabla f(y)\right]h + o(\|h\|),$$

which indicates that $\nabla g(y) := \nabla^2 f(y)(x - y) - \nabla f(y)$, from which we derive

$$\forall y \in \mathbb{R}^{d_\xi}, \quad \nabla^2 f(y)(x - y) = \nabla g(y) + \nabla f(y).$$

In particular,

$$\nabla^2 f(x_0)(x - x_0) = \nabla g(x_0) + \nabla f(x_0). \quad (13)$$

Plugging this into Eq. (12), we have

$$\begin{aligned}
f(x) &= f(x_0) + \nabla f(x_0)(x - x_0) + \frac{1}{2}\left[\nabla g(x_0) + \nabla f(x_0)\right](x - x_0) + o(\|x - x_0\|^2) \\
&= f(x_0) + \frac{3}{2}\underbrace{\nabla f(x_0)(x - x_0)}_{g(x_0)} + \frac{1}{2}\nabla g(x_0)(x - x_0) + o(\|x - x_0\|^2).
\end{aligned}$$

This concludes the proof. □

While this expression of the second-order Taylor expansion of a vector-valued function makes its implementation memory-efficient via Automatic Differentiation (AD), it still relies on nested derivatives, which scale exponentially with the order of the Taylor expansion due to avoidable recomputations. For even higher-order Taylor expansions, this scaling is frightfully inefficient. Taylor-Mode AD is a promising avenue to address this issue (Bettencourt et al., 2019).

## A.2  IDENTIFIABILITY OF PHYSICAL PARAMETERS

Identifying the underlying physical parameters of a system is of enduring interest to scientific machine learning practitioners. Our work critically builds on CAMEL (Blanke & Lelarge, 2024) which extensively studies a model similar to NCF-$t_1$ as a linearly parametrized system. Its Proposition

1 states, informally, that in the limit of vanishing training loss $\mathcal{L}(\cdot, \cdot, \cdot) = 0$, the relationship between the learned context vectors and the system parameters is linear and can be estimated using ordinary least square. This forces the model to learn a meaningful representation of the system instead of overfitting the examples from the training tasks.

We now reformulate Blanke & Lelarge (2024)'s identifiability result for linear systems trained with first-order Taylor expansion and the Random-All pool-filling strategy (see Appendix A.6), then we provide an alternative proof suited to our setting. Building on notations from Eqs. (1) and (4), we restate that our goal is to approximate the true vector field based on

$$\frac{\mathrm{d}x}{\mathrm{d}t}(t) = f_{\text{true}}(x(t), c), \qquad \text{and} \qquad \frac{\mathrm{d}x}{\mathrm{d}t}(t) = f_\theta(x(t), \xi), \qquad \forall t \in [0, T].$$

We drop the dependence on $x \in \mathbb{R}^d$ to ease notations in the sequel (the $\nabla$ will henceforth indicate gradients wrt $\xi$). As such, the predictor $f_\theta : \mathbb{R}^{d_\xi} \to \mathbb{R}^d$ is parametrized as a neural network and learned, while $f_{\text{true}} : \mathbb{R}^{d_c} \to \mathbb{R}^d$ is known and affine, i.e. $\exists P \in \mathbb{R}^{d \times d_c}$ and $p \in \mathbb{R}^d$ such that $f_{\text{true}}(c) = Pc + p$.

**Proposition 2** (Identifiability of affine systems). *Assume $d_\xi \geq d_c$, that $P$ is full-rank, and that $f_{\text{true}}$ is differentiable. In the limit of zero training loss in Eq. (10), $f_\theta$ is affine on an open region of $\mathbb{R}^{d_\xi}$. Furthermore, there exists $Q \in \mathbb{R}^{d_c \times d_\xi}$ and $q \in \mathbb{R}^{d_c}$ such that for any meta-trained $\xi \in \{\xi^e\}_{e=1}^m$ and its corresponding underlying parameter $c \in \{c^e\}_{e=1}^m$, we have $c = Q\xi + q$.*

*Proof.* Let $e \in [\![1, m]\!]$. In the limit of zero training loss, $f_\theta$ coincides with its first-order Taylor expansion in a neighborhood $U(\xi^e)$ which contains all other $\{\xi^j\}_{j=1}^m$. We can write

$$f_\theta(\xi) = f_\theta(\xi^e) + A(\xi - \xi^e), \qquad \forall \xi \in U(\xi^e) \tag{14}$$

where $A = \nabla f_\theta(\xi^e)$ is constant.

Similarly, for $j \in [\![1, m]\!]$, there exists an open set $U(\xi^j)$ which contains all other $\{\xi^e\}_{e=1}^m$, such that

$$f_\theta(\xi) = f_\theta(\xi^j) + B(\xi - \xi^j), \qquad \forall \xi \in U(\xi^j) \tag{15}$$

where $B = \nabla f_\theta(\xi^j)$ is constant.

To show that necessarily $A = B$, let's consider without loss of generality $\xi \in U(\xi^e) \cap U(\xi^j)$ (which is non-empty since both $\xi^e$ and $\xi^j$ are included in both sets) and proceed by contradiction, assuming $A \neq B$. Let $v \in \mathbb{R}^{d_\xi}$ sufficiently small, we can set $\xi = \xi^e + tv$ in Eq. (14) and write the directional derivative

$$\lim_{t \to 0} \frac{f_\theta(\xi^e + tv) - f_\theta(\xi^e)}{t} = \lim_{t \to 0} \frac{f_\theta(\xi^e) + A(\xi^e + tv - \xi^e) - f_\theta(\xi^e)}{t}$$
$$= \lim_{t \to 0} \frac{tAv}{t}$$
$$= Av. \tag{16}$$

We also write, using Eq. (15), the same directional derivative as

$$\lim_{t \to 0} \frac{f_\theta(\xi^e + tv) - f_\theta(\xi^e)}{t} = \lim_{t \to 0} \frac{f_\theta(\xi^j) + B(\xi^e + tv - \xi^j) - f_\theta(\xi^e)}{t}$$
$$= \lim_{t \to 0} \frac{f_\theta(\xi^j) - f_\theta(\xi^e) + B(\xi^e - \xi^j)}{t} + \lim_{t \to 0} \frac{tBv}{t}$$
$$= Bv. \tag{17}$$

For $v \notin \ker(B - A)$, we have $Av \neq Bv$ which contradicts with the uniqueness of directional derivatives for differentiable functions (Spivak, 1965). This shows that $f_\theta$ is affine on the open set $U = \bigcap_{e=1}^m U(\xi^e)$.

Furthermore, using Eq. (14), we have

$$f_\theta(\xi) - A\xi = f_\theta(\xi^e) - A\xi^e, \qquad \forall \xi \in U \tag{18}$$

which is valid for all $e \in [\![\,1, m\,]\!]$. This indicates that the right hand side of Eq. (18) is constant, i.e. $\exists \tilde{q} \in \mathbb{R}^{d_\xi}$ such that $\forall e \in [\![\,1, m\,]\!]$, $f_\theta(\xi^e) - A\xi^e = \tilde{q}$. We then use the fact that in the limit of zero training loss, the predicted and true vector fields coincide for a context $\xi \in U$ and its corresponding underlying parameters $c$ :

$$f_{\text{true}}(c) = f_\theta(\xi) \Rightarrow Pc + p = A(\xi - \xi^e) + f_\theta(\xi^e) \qquad \text{for } e \in [\![\,1, m\,]\!]$$
$$= A\xi + \tilde{q}.$$

Since $d_\xi \geq d_c$ and $P$ is full-rank, its rows are linearly independent, guaranteeing the existence of a pseudo-inverse. We can thus write $c = Q\xi + q$ with

$$Q = (P^T P)^{-1} P^T A, \qquad \text{and} \qquad q = (P^T P)^{-1} P^T [\tilde{q} - p]. \tag{19}$$

$\square$

The closed form expression Eq. (19) can be challenging to derive, especially since $P$ and $p$ might not be fully known when collecting data. So similar to (Blanke & Lelarge, 2024), one can perform post-training, ordinary least squares regression on observed $\{c^e\}_{e=1}^{m'}$ (with $m' \leq m$) to estimate the optimal $Q^*$ and $q^*$ :

$$Q^*, q^* \in \underset{Q,q}{\text{argmin}} \frac{1}{2} \sum_{e=1}^{m'} \|Q\xi^e + q - c^e\|_2^2. \tag{20}$$

**Experimental validation.** We validate Proposition 2 by modifying the Lotka-Volterra (LV) experiment. For CoDA (Kirchmeyer et al., 2022) and NCF-$t_1$, we use the exact same network architecture: a 4-layer MLP with 224 hidden units. This means no context nor state network is used in NCF-$t_1$: the context vector of size $d_\xi = 2$ is directly concatenated to the state vector as done by Zintgraf et al. (2019). We note that the results obtained using this configuration further indicate the superiority of NCF, even when model comparison centers on their main/root networks (see also (Park et al., 2023) for a similar model comparison based on parameter count)[5].

After meta-training and meta-testing, we set out to recover the underlying parameters of the Lotka-Volterra systems via a linear transformation of the learned context vectors. We fit a linear regression model to the 9 meta-training context vectors, using the true physical parameters as supervision signal. We test on the 4 adaptation contexts. The results, displayed in Fig. 5, adequately illustrate interpretability as stated in (Blanke & Lelarge, 2024, Proposition 1) and our Proposition 2. They show that our trained meta-parameters recover the underlying system parameters up to a linear transform, and thus enable **zero-shot** (physical parameter-induced) adaptation via inverse regression.

**Robustness to noise.** Additionally, system identification with NCFs is robust to noise in the trajectory. We show this empirically by corrupting the single trajectory in each adaptation environment with a Gaussian noise scaled by a factor of $\eta$. Upon addition of this noise, sequential adaptation is performed to recover new $\xi$ which are then transformed into $c$ and plotted. The weight $Q$ and bias $q$ of the affine transform are fitted on the training environments and their corresponding underlying parameters, which are unchanged across all noise levels. Fig. 7 shows that the reconstruction[6] MSE remains low despite the noise (compared to CoDA's MSE of $1.57 \times 10^{-2}$ when $\eta = 0$ shown in Fig. 5), and the physical system remains visually identifiable, especially in the convex hull of training environments. Outside the convex hull, the identifiability is notably worse with $\eta \geq 0.1$ indicating that this noise level is excessively high given the range of the LV state values.

### A.3 CONVERGENCE OF PROXIMAL ALTERNATING MINIMIZATION

For clarity of exposition, Theorem 1 expressing the convergence of Algorithm 1 to second-order stationary points is repeated below.

---

[5]As reported in Appendix D, the primary model comparison approach in this work counts all learnable parameters, including hypernetworks if involved. Only the context vectors are exempt from this count.

[6]Sample context vectors pre-reconstruction can be observed in Fig. 22 as the structure is preserved with changing seeds.

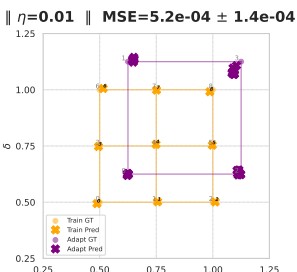 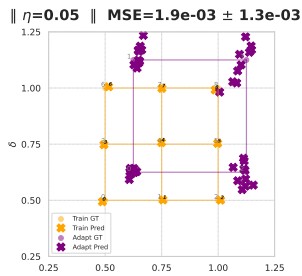 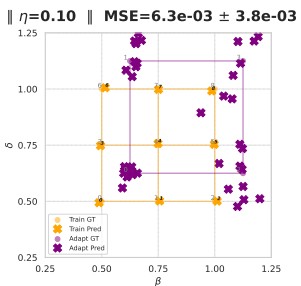

Figure 7: Robustness of NCF-$t_1$ when identifying physical parameters with varying levels of noise $\eta$ injected into the adaptation trajectory. The MSE is reported with the standard deviation across 10 runs with different seeds for the Gaussian noise. (Left) $\eta = 0.01$, (Middle) $\eta = 0.05$, (Right) $\eta = 0.1$.

**Theorem 1** (Convergence to second-order stationary points). *Assume that $\mathcal{L}(\cdot, \cdot, \mathcal{D}_{tr})$ satisfies the Kurdyka-Lojasiewicz (KL) property, is $L$ bi-smooth, and $\nabla \mathcal{L}(\cdot, \cdot, \mathcal{D}_{tr})$ is Lipschitz continuous on any bounded subset of domain $\mathbb{R}^{d_\theta} \times \mathbb{R}^{d_\xi \times m}$. Under those assumptions, let $(\theta_0, \xi_0^{1:m})$ be a random initialization and $(\theta_q, \xi_q^{1:m})$ be the sequence generated by Algorithm 1. If the sequence $(\theta_q, \xi_q^{1:m})$ is bounded, then it converges to a second-order stationary point of $\mathcal{L}(\cdot, \cdot, \mathcal{D}_{tr})$ almost surely.*

We use this section to briefly emphasize that the assumptions of KL property (Attouch et al., 2010) and Lipschitz continuity are mild and easily achievable with neural networks. Interestingly, boundedness of the sequence $(\theta_k, \xi_k^{1:m})$ is guaranteed a priory if $\mathcal{L}$ is coercive (Li et al., 2019), a property we encouraged by regularizing $\ell$ wrt weights and contexts as in Eq. (9).

### A.4 ORDINARY ALTERNATING MINIMIZATION

Here we provide an additional procedure (Algorithm 3) for training Neural Context Flows which we reserved for the NCF-$t_1$ variant. While stronger assumptions (Li et al., 2019) are needed to establish convergence guarantees like its proximal extension, is it relatively easier to implement, and exposes fewer hyperparameters to tune.

---

**Algorithm 3** Ordinary Alternating Minimization

1: **Input:** $\mathcal{D}_{tr} := \{\mathcal{D}_{tr}^e\}_{e \in [\![1,m]\!]}$
2: $\theta \in \mathbb{R}^{d_\theta}$ randomly initialized
3: $\xi^{1:m} := \bigcup_{e=1}^{m} \xi^e$, where $\xi^e = \mathbf{0} \in \mathbb{R}^{d_\xi}$
4: $\eta_\theta, \eta_\xi > 0$
5: **repeat**
6: $\quad \theta \leftarrow \theta - \eta_\theta \nabla_\theta \mathcal{L}(\theta, \xi^{1:m}, \mathcal{D}_{tr})$
7: $\quad \xi^{1:m} \leftarrow \xi^{1:m} - \eta_\xi \nabla_\xi \mathcal{L}(\theta, \xi^{1:m}, \mathcal{D}_{tr})$
8: **until** $(\theta, \xi^{1:m})$ converges

---

**Algorithm 4** Bulk Adaptation of NCF

1: **Input:** $\mathcal{D}_{ad} := \{\mathcal{D}^{e'}\}_{e' \in [\![a,b]\!]}$
2: $\theta \in \mathbb{R}^{d_\theta}$ learned
3: $\xi^{a:b} = \bigcup_{e'=a}^{b} \xi^{e'}$, where $\xi^{e'} := \mathbf{0} \in \mathbb{R}^{d_\xi}$
4: $\eta > 0$
5: **repeat**
6: $\quad \xi^{a:b} \leftarrow \xi^{a:b} - \eta \nabla_\xi \mathcal{L}(\theta, \xi^{a:b}, \mathcal{D}_{ad})$
7: **until** $\xi^{a:b}$ converges

---

### A.5 BULK ADAPTATION ALGORITHM

Neural Context Flows are designed to be fast at adaptation time. However, one might want to adapt to hundreds or thousands of environments, perhaps to identify where the performance degrades on downstream tasks. In such cases, Algorithm 2 could be slow due to its sequential nature. We provide Algorithm 4 that leverages the same parallelism exploited during training (while restricting information flow from one adaptation environment to the next).

Although highly parallelizable, we realize in practice that Algorithm 4 is susceptible to two pitfalls:

1. **Memory scarcity**: the operating system needs to allocate enough resources to store the data $\mathcal{D}_{ad}$, the combined context vectors $\xi^{a:b} \in \mathbb{R}^{d_\xi \times (b-a+1)}$ and backpropagate their gradients

in order to adapt all environments at once, which might be impossible if the context pool size $p$ is set too high. To avoid this issue, we recommend using $p \ll m$, or ideally $p = 1$ if contextual self-modulation is disabled.

2. **Slower convergence**: the bulk algorithm could take longer to converge to poorer context vectors when the jointly adapted environments are all very far apart. This is because the contextual self-modulation process would be rendered useless by such task discrepancy, and sampling $j = e$ in the context pool P to return to standard a Neural ODE in Section 3 would be harder. We find in practice that manually setting $j = e$ by adjusting the vector field to *forego the Taylor expansion* works well (see Fig. 3). A more direct way of achieving the same result is to disregard P and retain $f_\theta$ (rather than its Taylor expansion) in Eq. (5).

All adaptation results in this paper use the sequential adaptation procedure in Algorithm 2, except for the grid-wise adaptation in Section 4.

### A.6    What's in a context pool ?

The content of the context pool P not only defines the candidate trajectories we get from Eq. (8), but also the speed and memory cost of the meta-training process. Based on intuitive understanding of the role of P, we outline 3 tunable pool-filling strategies for selecting $p$ neighboring contexts:

- **Random-All (RA)**: all $p$ distinct contexts can be selected by randomly drawing their indices from $[\![\,1, m\,]\!]$. By repeatedly doing so, we maximize long-range interactions to provide the broadest form of self-modulation – since information can always (in the stochastic limit) flow from any environment into $e$.

- **Nearest-First (NF)**: only the $p$ closest contexts to $\xi^e$ are selected, thus encouraging environments to form clusters. (Note, however, that if $p = 1$, then P $= \{e\}$ itself, and no self-modulation occurs.) In one ablation study, we observe that this strategy is the most balanced with regard to training time and performance (see Appendix C.3).

- **Smallest-First (SF)**: the smallest contexts in $L^1$ norm are selected first. Since an environment with context close to $\mathbf{0}$ can be interpreted as an environment-agnostic feature like in (Kirchmeyer et al., 2022; Blanke & Lelarge, 2024), this strategy prioritizes the flow of information from that base or canonical environment to the one of interest $e$.

### A.7    Scalability of the NCF algorithms

The Neural Context Flow (NCF) framework demonstrates excellent scalability with respect to various hyperparameters, owing to its innovative design and implementation. This scalability is evident in three key aspects: distributed training capabilities, efficient handling of large context vectors, and utilization of first-order optimization techniques.

Primarily, NCF's training process can be distributed and parallelized across environments and trajectories (see Eq. (10), Fig. 12, and Algorithm 4), a feature that distinguishes it from baseline methods which are limited to parallelization across trajectories. Furthermore, the framework employs an efficient approach, as outlined in Proposition 3.1, to avoid materializing Jacobians or Hessians wrt potentially large context vectors, thereby significantly enhancing scalability. Additionally, NCF utilizes only first-order gradients wrt model weights $\theta$, in contrast to methods like CAVIA that require second-order information in their bi-level optimization loop.

Scalability can also be evaluated in terms of component size, particularly in meta-learning adaptation rules that incorporate additional contextual parameters beyond shared model weights. These components may include encoders, hypernetworks, and other mechanisms for generating or refining context vectors necessary for task adaptation. The complexity and memory requirements of contextual meta-learning rules, which are directly related to the size of these components, can be quantified by their parameter count among other metrics. In this regard, NCF maintains a constant memory cost $O(1)$, while baseline methods such as CAVIA and CoDA require additional memory to produce better contexts (see (Park et al., 2023, Table 1)).

Despite these advantages, the NCF framework may face challenges related to memory and computational efficiency due to the requirement of solving $p$ Neural ODEs in equation 3, as opposed to

a single one. In scenarios where all training environments are utilized (i.e., $p = m$), this results in a quadratic cost $O(m^2)$ for Algorithms 1 and 3. However, our ablation studies in Appendix C.1 demonstrate that competitive performance can be achieved on most problems using as few as $p = 2$ neighboring environments. Additional studies in Appendices C.2 and C.4 establish the necessity of expressive context vectors (Voynov & Babenko, 2020) and validate the efficacy of the 3-networks architecture, respectively.

It is worth noting that limiting these quantities could directly contribute to improved parameter counts and more interpretable models. Moreover, restricting the total number of environments contributing to the loss equation 10 at each iteration may further enhance efficiency.

## B  DATASETS & ADDITIONAL RESULTS

### B.1  GEN-DYNAMICS

Given the lack of benchmark consistency in Scientific Machine Learning (Massaroli et al., 2020), we launched `Gen-Dynamics`: `https://github.com/ddrous/gen-dynamics`. This is a call for fellow authors to upload their metrics and datasets, synthetic or otherwise, while following a consistent interface. In the context of OoD generalization for instance, we suggest the dataset be split in 4 parts: • (1) `train`: For In-Domain meta-training; • (2) `test`: For In-Domain evaluation; • (3) `ood_train`: For Out-of-Distribution adaptation to new environments (meta-testing); • (4) `ood_test`: For OoD evaluation.

Each split should contain trajectories `X` and the time points `t` at which the states were recorded. The time `t` is a 1-dimensional array, and we recommend a 4-dimensional `X` tensor with dimensions described as follows: • (1) `nb_envs`: Number of distinct environments; • (2) `nb_trajs_per_env`: Number of trajectories per environment; • (3) `nb_steps_per_traj`: Number of time steps per trajectory (matching the size of `t`); • (4) `state_size`: Size of the state space.

While the suggestions apply mostly to dynamical systems' Meta-Learning, we believe they are generalizable to other problems, and are open to suggestions from the community. All problems described below are now represented in `Gen-Dynamics`.

Finally, we define the MSE and the mean absolute percentage error (MAPE) criteria as they apply to all trajectory data found in `Gen-Dynamics`. Unless stated otherwise, the following are the metrics used throughout this work, including in Table 2.

$$\text{MSE}(x, \hat{x}) = \frac{1}{N \times d} \sum_{n=1}^{N} \|x(t_n) - \hat{x}(t_n)\|_2^2, \tag{21}$$

$$\text{MAPE}(x, \hat{x}) = \frac{1}{N \times d} \sum_{n=1}^{N} \left| \frac{x(t_n) - \hat{x}(t_n)}{x(t_n)} \right| \times 100. \tag{22}$$

### B.2  UNCERTAINTY ESTIMATION

Neural Context Flows can provide uncertainty about their predictions. To show this, we calculate (*i*) the relative mean squared error (MSE), (*ii*) the mean absolute percentage error (MAPE), and (*iii*) the 3-$\sigma$ Coverage or Confidence Level (CL) (Serrano et al., 2024), with the following formulae applied In-Domain:

$$\text{Rel. MSE} = \frac{100}{m \times S \times N \times d} \sum_{e=1}^{m} \sum_{i=1}^{S} \sum_{n=1}^{N} \frac{\|x_i^e(t_n) - \hat{\mu}_i^e(t_n)\|_2^2}{\|x_i^e(t_n)\|_2^2}, \tag{23}$$

$$\text{MAPE} = \frac{100}{m \times S \times N \times d} \sum_{e=1}^{m} \sum_{i=1}^{S} \sum_{n=1}^{N} \sum_{d'=1}^{d} \left| \frac{x_{i,d'}^e(t_n) - \hat{\mu}_{i,d'}^e(t_n)}{x_{i,d'}^e(t_n)} \right|, \tag{24}$$

$$\text{CL} = \frac{100}{m \times S \times N \times d} \sum_{e=1}^{m} \sum_{i=1}^{S} \sum_{n=1}^{N} \sum_{d'=1}^{d} \mathbb{1}_{x_{i,d'}^e(t_n) \in \text{CI}(e,i,n,d')}, \tag{25}$$

with the mean and standard deviation across candidate trajectory predictions defined as

$$\hat{\mu}_i^e(t_n) = \frac{1}{p} \sum_{j=1}^{p} \hat{x}_i^{e,j}(t_n), \quad \text{and } \hat{\sigma}_i^e(t_n) = \sqrt{\frac{1}{N} \sum_{j=1}^{p} (\hat{x}_i^{e,j}(t_n) - \hat{\mu}_i^e(t_n))^2}, \tag{26}$$

and the pointwise empirical confidence interval defined as

$$\text{CI}(e, i, \cdot, \cdot) = [\hat{\mu}_i^e - 3\hat{\sigma}_i^e, \hat{\mu}_i^e + 3\hat{\sigma}_i^e], \tag{27}$$

where $S$ indicates the number of trajectories used per environment, $N$ the length of each trajectory, and $d$ the dimensionality of the problem. The number of InD environments is denoted by $m$, to be replaced with $b - a + 1$ for OoD cases.[7] We set $p = m$ for InD uncertainty metrics calculation,

---

[7]Consequently, the first summation symbol's bounds and its corresponding factor in the denominators of Eqs. (23) to (25) should be adjusted accordingly for OoD formulae.

and we are guaranteed the existence of those same $m$ environments for OoD cases. However, if our model performs well across all adaptation environments it encounters –as is the case with NCF-$t_2$ as observed in Table 2– then *all* training and adaptation environments can be used, resulting in $p = m + b - a + 1$ instead of only $p = m$ environments at our disposal for Eq. (26). This produces more sample predictions, allowing for more reliable population statistics.

The results in Table 3 show low relative MSE on ODE problems, and remarkably low MAPE scores on all problems. These indicate that the empirical mean of the predictions is indeed close to the ground truth. We also notice that the confidence levels vary from very low values on NS to very high on BT. Based on Eq. (25), we hypothesize that this is primarily due to standard deviations across predictions.[8]

To test our hypothesis, we plot the standard deviations as they evolve with time in Fig. 8. When compared to per-problem InD and OoD CLs from Table 3, Fig. 8 reveals that higher confidence levels align with higher standard deviations, which is particularly noticeable on forecasts beyond training time horizons. Additionally, in Fig. 9, we plot the pointwise standard deviations as they relate to the corresponding absolute errors. Naturally, we observe a non-negligible correlation of the two, especially on the GO problem. Focusing on OoD behavior, our model successfully avoids undesirable regions of low uncertainty but higher-than-InD errors (along the lop left corners). Instead, some OoD predictions for ODE problems fall in a region of higher-than-InD uncertainty, but still low error (along the bottom right corners), which stresses the well-suitedness of our approach for OoD generalization. Despite not knowing the underlying source of uncertainty we wish to model, these results suggest that our framework is capable of providing meaningful uncertainty estimates. This said, we emphasize that the calculation and interpretation of aforementioned uncertainty metrics (e.g. the width of CI) should be grounded on knowledge and goals of the problem at hand.

Table 3: Uncertainty estimation metrics with NCF-$t_2$, all expressed in percentage points (%). The star $*$ indicates cases where the close to zero denominators had to be filtered out to retain state values greater than $10^{-3}$. The Relative MSE is the most sensitive to these instabilities due to squaring at the denominator.

| | LV | | | GO | | |
|---|---|---|---|---|---|---|
| | REL. MSE | MAPE | CL | REL. MSE | MAPE | CL |
| IND | 0.032 | 0.80 | 56.22 | 5.119 | 10.36 | 82.06 |
| OOD | 0.183 | 1.86 | 78.24 | 1.653 | 7.17 | 70.47 |
| | SM | | | BT | | |
| | REL. MSE | MAPE | CL | REL. MSE | MAPE | CL |
| IND | 2.005* | 4.07* | 65.64 | 42.199 | 18.95 | 92.25 |
| OOD | 0.158* | 1.99* | 94.70 | 46.028 | 22.28 | 90.63 |
| | GS | | | NS | | |
| | REL. MSE | MAPE | CL | REL. MSE | MAPE | CL |
| IND | 2118.51* | 58.33* | 14.58 | 166.696* | 17.30* | 11.19 |
| OOD | 281.986* | 33.43* | 11.89 | 152.78* | 16.72* | 11.09 |

### B.3 SIMPLE PENDULUM (SP)

The autonomous dynamical system at play here corresponds to a frictionless pendulum suspended from a stationary point by a string of fixed length $L$. The state space $x = (\alpha, \omega)$, comprises the angle the pendulum makes with the vertical, and its angular velocity, respectively

$$\begin{cases} \dfrac{\mathrm{d}\alpha}{\mathrm{d}t} = \omega, \\ \dfrac{\mathrm{d}\omega}{\mathrm{d}t} = -\dfrac{g}{L}\sin(\alpha). \end{cases} \quad (28)$$

---

[8]We remark that the empirical confidence level CL is a metric that favors models that are uncertain in giving the right mean prediction.

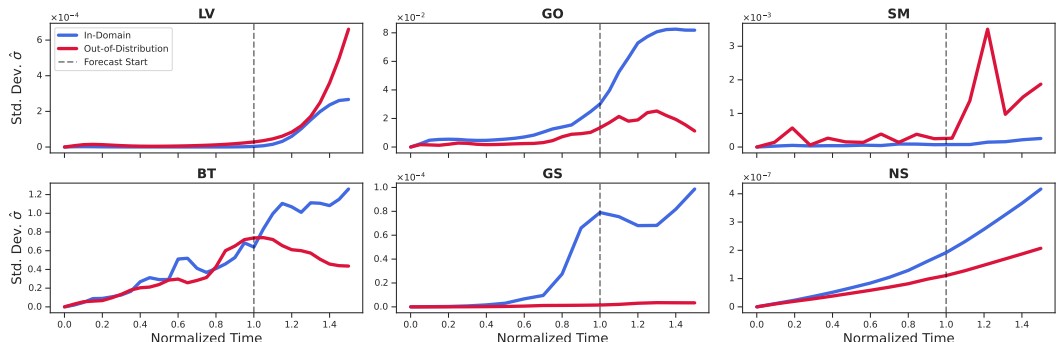

Figure 8: Average standard deviations indicating how uncertainty grows with time, including when the model forecasts in time domains not seen during training. Higher standard deviations correlate with higher confidence level metrics observed in Table 3.

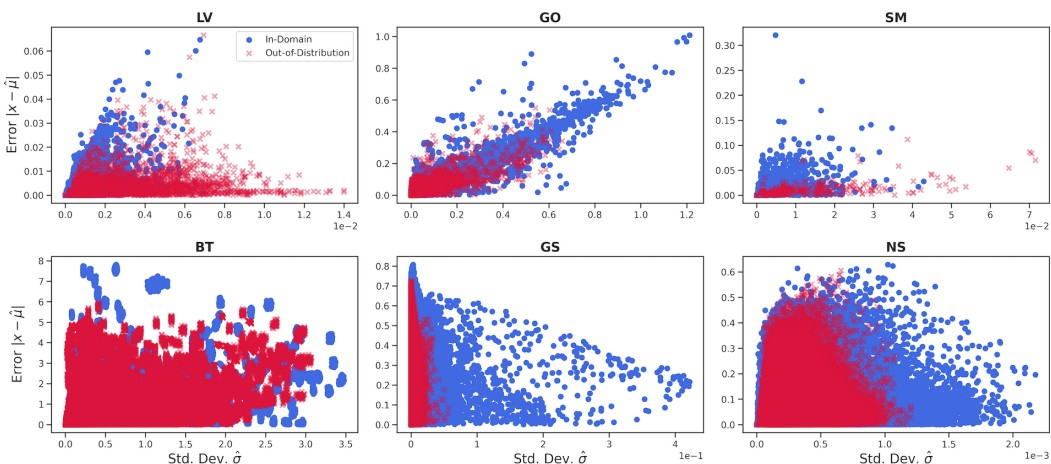

Figure 9: Pointwise absolute errors against standard deviations for all problems.

For this problem, each environment corresponds to a different gravity.[9] With $L = 1$ set, the goal is to learn a dynamical system that easily generalizes across the unobserved $g$. Trajectories are generated with a Runge-Kutta 4th order solver, and a fixed step size of $\Delta t = 0.25$. Only the NCF-$t_1$ variant is used in all experiments involving this problem.

**OFA vs. OPE vs. NCF.** The training and adaptation MSE metrics were reported in Table 1 during our favorable comparison to two baselines: One-For-All (OFA) – one context-free model trained for all environments; and One-Per-Env (OPE) – one model trained from scratch for each environment we encounter.[10] Here, we expand on said comparison with our loss values during training reported in Fig. 10. Additionally, we probe the training time of the different methods. Given that OFA and OPE do not require contextual information, we increase the capacity of their main networks to match the NCF total parameter count. Each method is then trained until the loss stagnates, and we report the amortized training times in Table 4.

While exhibiting much larger adaptation times, OPE overfits to its few-shot trajectories as evidenced in Fig. 10. The same figure shows that the OFA training loss quickly stagnates to a relatively high

---

[9]To give an intuition behind the term "environment", one might consider the surface of a celestial body in the solar system (see Section 1).

[10]The OFA paradigm offers no mechanism to adapt to new environments, unless we fine-tune the vector field's weights, thus returning to an OPE-like setting.

value during training. Unsurprisingly, learning one context-agnostic vector field for all environments (OFA) is suboptimal given the vast differences in gravity from one environment to the next. These observations align with the more complete study on OFA and OPE by Yin et al. (2021a). Leveraging Meta-Learning, our method, trained on all environments at once like OFA, effectively learns to discriminate between them and produces low validation MSE metrics and accurate trajectories, one of which is presented in Fig. 1b. Taken together, Tables 1 and 4 and Fig. 10 show that Meta-Learning delivers on training time, adaptation time, and most importantly, testing accuracy.

Table 4: Meta-training and adaptation times ($\downarrow$) with OFA, OPE, and NCF-$t_1$ on the SP problem. We report the amortized times (in **minutes**) corresponding to fitting one single environment (of which we count 25 when meta-training, and 2 for adaptation).

|      | #PARAMS | TRAIN | ADAPT |
|------|---------|-------|-------|
| OFA  | 49776   | 0.34  | 0     |
| OPE  | 49776   | 4.63  | 5.72  |
| NCF  | 50000   | 2.96  | 0.51  |

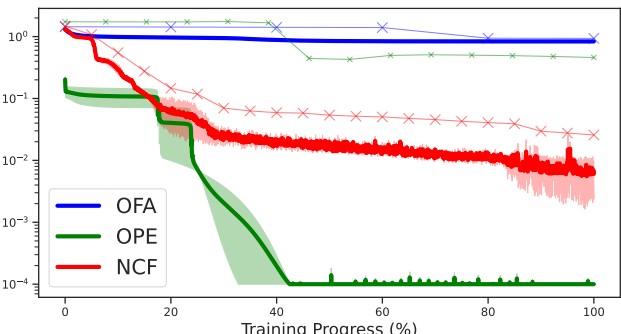

Figure 10: MSE loss values when training NCF-$t_1$ on the SP problem, compared with the baseline OFA and OPE formulations. The crosses $\times$ indicate mean validation curves across 3 runs, color-coded to match the training curves. OFA fails catastrophically since the diversity of environments in the training dataset prevents the approximation of any meaningful vector field, while OPE overfits to its 4 training trajectories.

**Sample efficiency with the number of trajectories.** We compare our model to the two OFA and OPE baselines as the number of trajectories $S$ in each training environment is increased from 1 to 12. The results reported in Fig. 11 indicate that NCF is indeed the best option when data is limited, while the improvements in OFA MSEs are barely noticeable. As $S$ increses, we observe that OPE is able to overcome overfitting to ultimately achieve the best results. These results demonstrate that NCF effiently uses its few-shots trajectories. However, if neither data nor training time are not a concern (cf. Table 4), then the traditional One-Per-Env should be prioritized.

**Scaling with the training environments.** The computational speed of any method based on Neural ODEs depends on the numerical integrator it uses. To provide consistent number of function evaluations (NFEs), we switch the adaptive time-stepper `Dopri5` for the fixed time-stepper `RK4`, then we measure the duration of epochs as the training progresses. These times as used to produce Fig. 12, which indicates training times per epoch (in seconds) as the number of training environments is increased while keeping the range of gravities unchanged between 2 and 24. We observe excellent scaling, with the training time only increasing by roughly 23% (from 0.38 to 0.47 seconds) when the number of environments is scaled by 10 (from 5 to 50).

**Probing the context vectors.** Beyond serving as a control signal for the vector fields, the contexts encode useful representations. In Fig. 13, we visualize the first two dimensions of the various

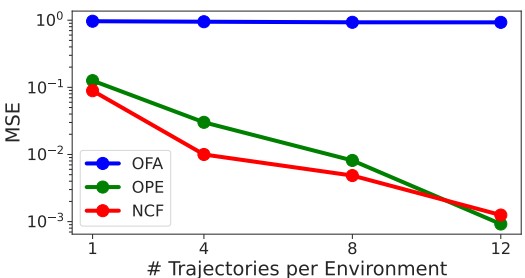

Figure 11: In-Domain MSEs on the SP problem comparing the sample eficiency of NCF-$t_1$ against OPE and OFA. NCF is effective in low-data regimes, and OPE overcomes the gap as data increases.

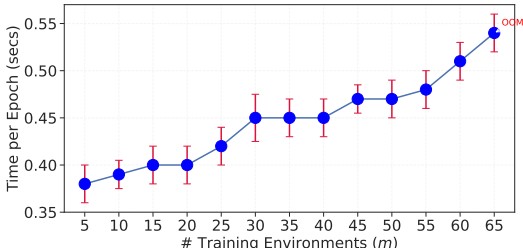

Figure 12: Training time as the number of training environments $m$ is increased from 5 to 65. The vertical bars are proportional to the standard deviation across epochs. OOM indicates that our workstation ran out of memory.

$\{\xi^e\}_{1 \le e \le 25}$ after training. We observe that environments close in indices[11] are equally close in the two context dimensions. Similarly, distant environments are noticeably far apart in this view of the context space. The same observation is made during adaptation, where, for instance, $e' = a_2$ (corresponding to $g = 14.75$) gets a context close to $e = 15$ (corresponding to $g = 14.83$). This observation indicates that the latent context vector is encoding features related to gravity, which may be used for further downstream representation learning tasks.

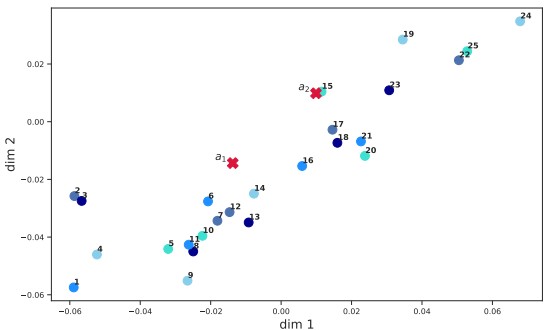

Figure 13: Representation of the first and second dimensions of the learned contexts for the SP problem. The labels 1 to 25 identify the training environments (in shades of blue), while $a_1$ and $a_2$ (in red), indicate the adaptation environments. We observe that environments close in indices (and thus in gravity) share similar contexts along these dimensions.

---

[11]The indices of the training environments correspond to their ordering in increasing values of gravity.

## B.4 LOTKA-VOLTERRA (LV)

With dynamics that are of continued interest to many fields (epidemiology (Venturino, 1994), economy (Wu et al., 2012), etc.), the Lotka-Volterra (LV) ODE models the evolution of the concentration of preys $x$ and predators $y$ in a closed ecosystem. The behavior of the system is controlled by the prey's natural growth rate $\alpha$, the predation rate $\beta$, the predator's increase rate upon consuming prey $\delta$, and the predator's natural death rate $\gamma$

$$\begin{cases} \dfrac{\mathrm{d}x}{\mathrm{d}t} & = \alpha x - \beta xy, \\ \dfrac{\mathrm{d}y}{\mathrm{d}t} & = \delta xy - \gamma y. \end{cases} \tag{29}$$

We repeat the experiment as designed in (Kirchmeyer et al., 2022). All synthetic ground truth data is generated with the two initial states both following the $\mathcal{U}(1,3)$ distribution. Once sampled, we note that the same initial condition is used to generate trajectories for all environments. The parameters that vary across training environments are $\beta \in \{0.5, 0.75, 1\}$ and $\delta \in \{0.5, 0.75, 1\}$. In each training environment, we generate 4 trajectories with a Runge-Kutta time-adaptive 4th-order scheme, while we generate 32 for In-Domain evaluation. For one-shot adaptation, we extrapolate to $\beta \in \{0.625, 1.125\}$ and $\delta \in \{0.625, 1.125\}$, with only 1 trajectory per environment, and 32 for OoD testing. The observed parameters $\alpha$ and $\gamma$ are always fixed at 0.5.

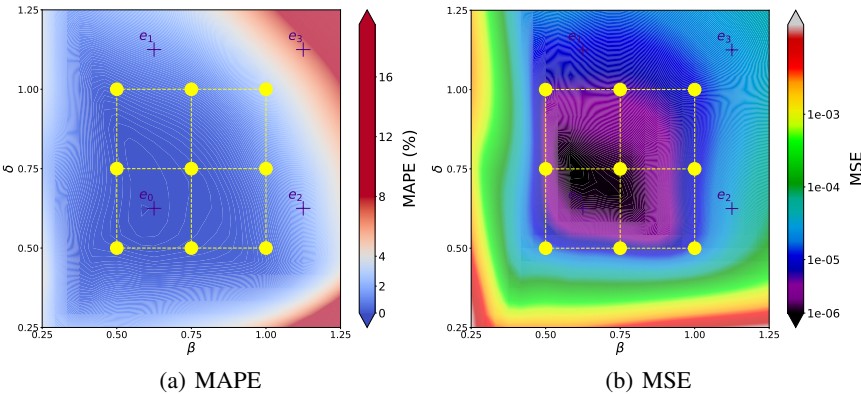

(a) MAPE  (b) MSE

Figure 14: Results for the Large adaptation of the LV problem to a grid, showcasing the MAPE and training MSE. Compared to Fig. 3, the colorbar range in (a) is shrunk to focus on the MAPEs between 0 and 8%.

The large grid-wise adaptation experiment conducted on the LV problem in Section 4.3 showed that NCFs were powerful at extrapolating. Here, through the MSE training losses in Fig. 14(b), we highlight the fact that the bottom and left edges of the grid display quite high errors. This illustrates the importance of observing several loss metrics when learning time series (Hewamalage et al., 2023).

### B.5 GLYCOLYTIC-OSCILLATOR (GO)

This equation defines the evolution of the concentration of 7 biochemical species $\{s_i\}_{i=1,2,...,7}$ according to the ODE:

$$\frac{ds_1}{dt} = J_0 - k_1 \frac{s_1 s_6}{1 + (s_6/K_1)^q}$$

$$\frac{ds_2}{dt} = 2k_1 \frac{s_1 s_6}{1 + (s_6/K_1)^q} - k_2 s_2(N - s_5) - k_6 s_2 s_5$$

$$\frac{ds_3}{dt} = k_2 s_2(N - s_5) - k_3 s_3(A - s_6)$$

$$\frac{ds_4}{dt} = k_3 s_3(A - s_6) - k_4 s_4 s_5 - \kappa(s_4 - s_7)$$

$$\frac{ds_5}{dt} = k_2 s_2(N - s_5) - k_4 s_4 s_5 - k_6 s_2 s_5$$

$$\frac{ds_6}{dt} = -2k_1 \frac{s_1 s_6}{1 + (s_6/K_1)^q} + 2k_3 s_3(A - s_6) - k_5 s_6$$

$$\frac{ds_7}{dt} = \psi\kappa(s_4 - s_7) - k s_7$$

where the parameters either vary or are set fixed as per Table 5.

Table 5: Physical parameters and their values for the GO problem

| Parameter | $J_0$ | $k_1$ | $k_2$ | $k_3$ | $k_4$ | $k_5$ | $k_6$ | $K_1$ | $q$ | $N$ | $A$ | $\kappa$ | $\psi$ | $k$ |
|---|---|---|---|---|---|---|---|---|---|---|---|---|---|---|
| Value | 2.5 | (varies) | 6 | 16 | 100 | 1.28 | 12 | (varies) | 4 | 1 | 4 | 13 | 0.1 | 1.8 |

Again, we follow the same procedure as in (Kirchmeyer et al., 2022) to generate trajectories for each environment. Namely, we sample initial conditions from a specific distribution (Daniels & Nemenman, 2015, Table 2). We vary $k_1 \in \{100, 90, 80\}$ and $K_1 \in \{1, 0.75, 0.5\}$ to create 9 training environments with 32 trajectories each, both for InD training and testing. We use 1 trajectory for OoD adaptation to 4 environments defined by $k_1 \in \{85, 95\}$ and $K_1 \in \{0.625, 0.875\}$. We use 32 trajectories for OoD evaluation.

Intuitive post-processing of the contexts offers many insights into the meta-training process, particularly because it clusters the 9 environments into groups of 3, as illustrated in Fig. 15.

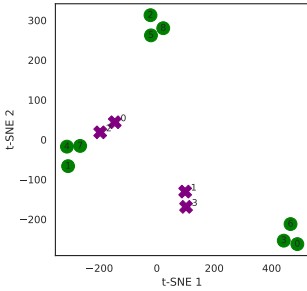

Figure 15: Illustration of 256-dimensional context vectors consistently clustered with t-SNE embeddings with perplexity of 2 on the Glycolytic Oscillator (GO). The training environments are in green and labeled as their IDs, while the adaptation environments are in purple. This clustering mirrors the distribution of the system parameters.

### B.6 SEL'KOV MODEL (SM)

We introduce the Sel'kov model in dimensionless form (Strogatz, 2018), a highly non-linear ODE mainly studied for its application to yeast glycolysis :

$$\frac{dx}{dt} = -x + ay + x^2y$$

$$\frac{dy}{dt} = \textcolor{red}{b} - ay - x^2y$$

where $a = 0.1$ and $b$ is the parameter that changes. The trajectories are generated with a time horizon of 40, with 11 regularly spaced time steps. We sample each initial condition state from the distribution $\mathcal{U}\{0, 3\}$, and we observe the appearance of a limit cycle (L1), then an equilibrium point (E), and another limit cycle as $b$ changes (see Fig. 4(a)).

Specifically, our 21 training environments are a union of 7 environments evenly distributed in $b \in [-1, -0.25]$, then 7 evenly distributed in $b \in [-0.1, -0.1]$, and finally 7 others with $b \in [0.25, 1]$. We generate 4 trajectories for training and 4 for InD testing. As for adaptation, we choose 6 environments, with $b \in \{-1.25, -0.65, -0.05, 0.02, 0.6, 1.2\}$ (see Fig. 16). We set aside 1 trajectory for adaptation and 4 for OoD testing.

In Section 4, we showed that NCF-$t_2$ outperformed other adaptation rules. We note, however, that training such systems is very difficult, even with NCFs. We observed in practice that convergence is extremely sensitive to weight initialization, as the shaded regions on the loss curves of Fig. 4(b) attest. The curves are complemented with Fig. 17, emphasizing that the model performs best in environments near the equilibrium.

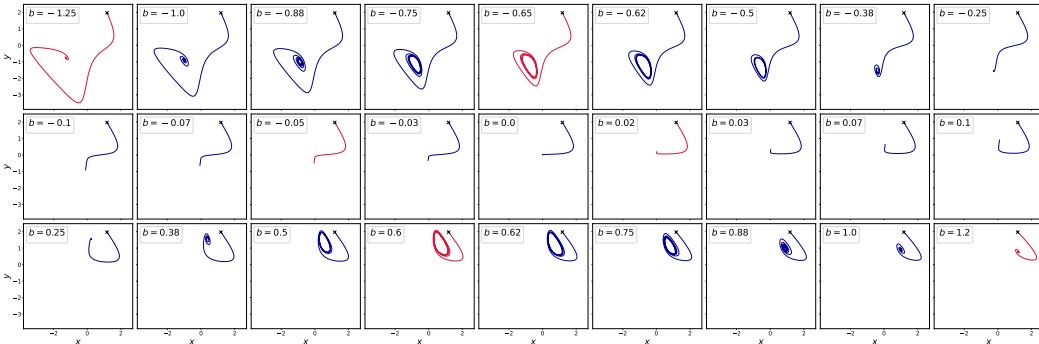

Figure 16: Visualization of a trajectory with the same initial condition in various environments of the Sel'kov Model's dataset. In blue are the meta-training environments, in red the meta-testing ones. We observe three attractors: the limit cycle (L1) along the top row, the fixed equilibrium (E) along the second row, and another limit cycle (L2) along the third row.

### B.7 BRUSSELATOR (BT)

The Brusselator model (Prigogine & Lefever, 1968) describes the reaction-diffusion dynamics of two chemical species, $U$ and $V$, and is given by the following system of partial differential equations (PDEs) defined on a $8 \times 8$ grid:

$$\frac{\partial U}{\partial t} = D_u \Delta U + \textcolor{red}{A} - (\textcolor{red}{B} + 1)U + U^2V,$$

$$\frac{\partial V}{\partial t} = D_v \Delta V + \textcolor{red}{B}U - U^2V,$$

where:

- $U$ and $V$ are the concentrations of the chemical reactants.
- $D_u = 1$ and $D_v = 0.1$ are the diffusion coefficients for $U$ and $V$, respectively.

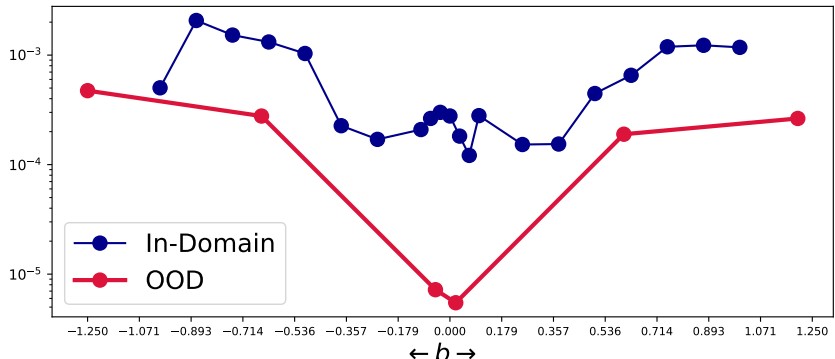

Figure 17: Per-environment In-Domain and adaptatation MSEs for the SM problem. In blue are the meta-training environments, in red the meta-testing ones. Central environments in the attractor (E) are better resolved than the others.

- $A$ and $B$ are constants representing the rate parameters of the chemical reactions, both varying across environments.

Using an RK4 adaptive time-step solver, the system is simulated up to $T = 10$ (excluded), with the step reported every time step $\Delta t = 0.5$. The initial condition for each trajectory for all environments is sampled and broadcasted as follows:

$$U_0 = \bar{A}$$

$$V_0 = \frac{\bar{B}}{\bar{A}} + 0.1\eta_{ij}$$

where $\bar{A} \sim \mathcal{U}(0.5, 2.0)$, $\bar{B} \sim \mathcal{U}(1.25, 5.0)$, and $\eta_{ij} \sim \mathcal{N}(0, 1)$ for each grid position $(i, j)$.

We aim to keep all environments involved in this problem outside the Brusselator's oscillatory regime $B^2 > 1 + A^2$. For Meta-training, $A$ and $B$ are selected from $\{0.75, 1, 1.25\} \times \{3.25, 3.5, 3.75\}$ yielding 12 environments, each with 4 trajectories for training, and 32 for InD testing. For adaptation, we select 12 environments from $\{0.875, 1.125, 1.375\} \times \{3.125, 3.375, 3.625, 3.875\}$, with 1 trajectory for training, and 32 for OoD testing.

As observed with the metrics in Table 2, the BT dataset is one of the most challenging to learn on. Indeed, the non-Meta-Learning baselines OFA and OPE equally struggle on it. To show this, we vary the number of trajectories in each trainnig environment between 1 and 8. Then, we plot in Fig. 18 the sample efficiency of our NCF approach against the non-Meta-Learning baselines. Like with the SP problem (cf. Fig. 11), we observe good NCF performance when data is scarce, underlining the suitability of our approach for few-shot learning.

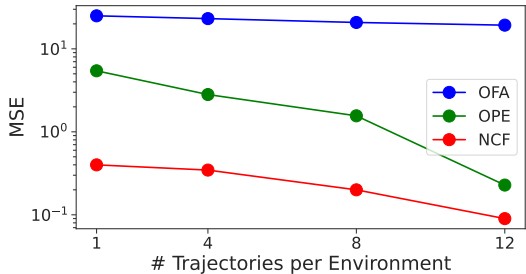

Figure 18: In-Domain MSEs on the BT problem comparing the sample eficiency of NCF-$t_2$ against OPE and OFA. NCF is effective in low-data regimes, and OPE closes the gap as data increases.

## B.8 GRAY-SCOTT (GS)

We aim to validate the empirical potential of our method on spatiotemporal systems beyond ODEs. We thus identify the Gray-Scott (GS) model for reaction-diffusion, a partial differential equation (PDE) defined over a spatial $32 \times 32$ grid:

$$\frac{\partial U}{\partial t} = D_u \Delta U - UV^2 + F(1 - U)$$

$$\frac{\partial V}{\partial t} = D_v \Delta V + UV^2 - (F + k)V$$

Like in the Brusselator case above, $U$ and $V$ represent the concentrations of the two chemical components in the spatial domain with periodic boundary conditions. $D_u, D_v$ denote their diffusion coefficients respectively, while $F$ and $k$ are the reaction parameters. We generate trajectories on a temporal grid with $\Delta t = 40$ and temporal horizon $T = 400$.

The parameters we use to generate our environments are the reaction parameters $F$ and $k$. We consider 4 environments for meta-training: $F \in \{0.30, 0.39\}, k \in \{0.058, 0.062\}$; and 4 others for adaption: $F \in \{0.33, 0.36\}, k \in \{0.59, 0.61\}$. Other simulation parameters, as well as the initial condition-generating distribution, are inherited from (Kirchmeyer et al., 2022), where we direct readers for additional details.

## B.9 NAVIER-STOKES (NS)

Like the Gray-Scott case, the 2D incompressible Navier-Stokes case is inherited from (Kirchmeyer et al., 2022). The PDE is defined on a $32 \times 32$ spatial grid as:

$$\frac{\partial \omega}{\partial t} = -v \nabla \omega + \nu \Delta \omega + f$$

$$\nabla v = 0$$

where:

- $\omega = \nabla \times v$ is the vorticity,
- $v$ is the velocity field,
- $\nu$ is the kinematic viscosity.

The trajectory data from $t = 0$ to $T = 10$ with $\Delta t = 1$ is obtained through a custom Euler integration scheme. By varying the viscosity from $\nu \in \{8 \cdot 10^{-4}, 9 \cdot 10^{-4}, 1.0 \cdot 10^{-3}, 1.1 \cdot 10^{-3}, 1.2 \cdot 10^{-3}\}$ we gather 5 meta-training environments, each with 16 trajectories for training and 32 for testing. Similarly, we collect 4 adaptation environments with $\nu \in \{8.5 \cdot 10^{-4}, 9.5 \cdot 10^{-4}, 1.05 \cdot 10^{-3}, 1.15 \cdot 10^{-3}$ each with 1 trajectory for training, and 32 for testing. Other parameters of the simulation, as well as the initial condition generating distribution, are inherited from (Kirchmeyer et al., 2022), where we encourage the readers to find more details.

# C ABLATION STUDIES

The ablation studies described in this section are designed to investigate how the context pool size, the context size, the pool-filling strategy, and the 3-networks architecture affect the performance of NCFs.

## C.1 LIMITING THE CONTEXT POOL SIZE

The context pool P from which environments $j$ are randomly sampled contributes significantly to the computational and memory complexity of the algorithm. As evidenced in Algorithms 1 and 3, the computation of the loss and hence its gradient can be parallelized across the contexts vectors $\xi^j$. With the goal of assessing the associated computational burden, we vary the size of the pool size for the LV problem from 1 to 9, reporting the In-Domain and OoD metrics, and computational time in Fig. 19.

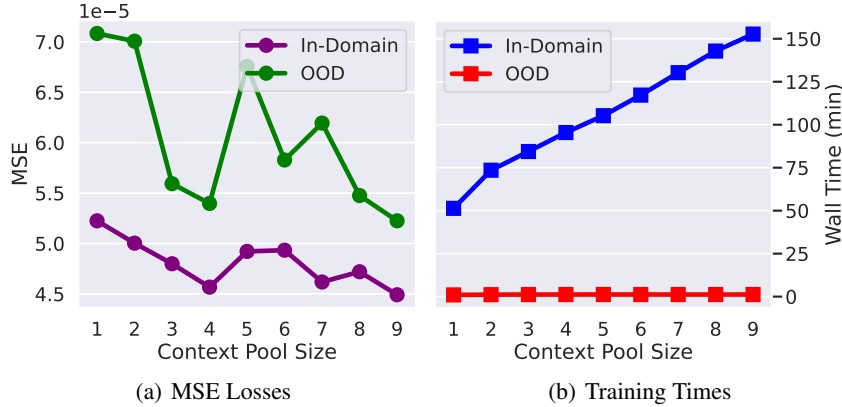

(a) MSE Losses          (b) Training Times

Figure 19: (a) In-domain and OoD MSEs, and (b) training and adaptation times when varying the context pool size on the LV problem with NCF-$t_2$.

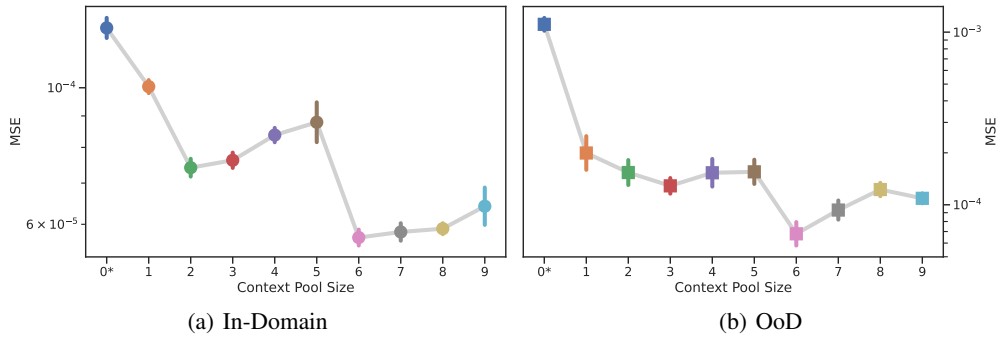

(a) In-Domain          (b) OoD

Figure 20: Average MSEs for the ablation of the context pool size for NCF-$t_1$, with evaluation carried over many seeds, hence the vertical bars indicating the standard deviations.

We dig further into the influence of the context pool size. While its ideal value might be difficult to conclude based on Fig. 19, Fig. 20 investigates NCF-$t_1$ to highlight how important any value bigger than 1 is (with $p = 6$ proving to be exceptionally adequate). Indeed, 0* indicates no context pool was used, and the vector field is evaluated without Taylor expansion. This ablation results in a significant drop in MSE of about one order of magnitude. While computational meta-training time scales linearly with the pool size (see Fig. 19(b)), the various losses do not. Overall, these results suggest that the context pool size $p$ largely remains a hyperparameter that should be tuned for maximum balance of accuracy and training time.

Our motivations for the above conclusion lies in that Taylor expansion is the key to information flowing from one environment to another, along with the concept of "task relatedness" or "context proximity". Indeed, depending on the pool-filling strategy used, some environments that are prohibitively far from one another may be required to interact in context space. As a result, the Taylor approximation incurs a non-reducible residual error term. To counter this effect, one could restrict the pool size to limit the impact of those far-apart environments since they will be less often sampled from the pool as the training progresses.

### C.2 RESTRICTION OF THE CONTEXT SIZE

Rather than limiting the pool size, what if the context vectors themselves were limited? The latent context vectors are the building blocks of NCF, and having shown that they encode useful representations vital for downstream tasks in Appendix B.3, we now inquire as to how their size influences the overall learning performance. This further provides the opportunity to test the contextual self-modulation process. Indeed, over-parametrization should not degrade the performance of NCFs since context vectors must be automatically kept small and close to each other (in $L^1$ norm) for the Taylor approximations to be accurate.

Like in CoDA (Kirchmeyer et al., 2022), the context size $d_\xi$ is directly related to the parameter count of the model; and limiting the parameter count bears practical importance for computational efficiency and interpretability. Thus, we perform the LV experiment as described in Appendix B.4 with $d_\xi \in \{2, 4, 8, 16, 32, 64, 128, 256, 512, 1024\}$. The results observed in Fig. 21(a) align with our intuitive understanding of increased expressiveness with bigger latent vectors. Together, Figs. 21(a) and 21(b) shed light on the relationship between $\xi$ and the underlying physical parameters pair $(\beta, \delta)$, suggesting that NCFs would benefit from the vast body of research in representation learning.

We appreciate that while selecting $d_\xi$ small is more interpretable (as can be observed by the clustering of the training environment Fig. 21(b)), it is important to choose $d_\xi$ sufficiently big. Indeed, using the JVP-based implementation we provide in Appendix E, large context vectors come at a reduced extra cost in both speed and memory.

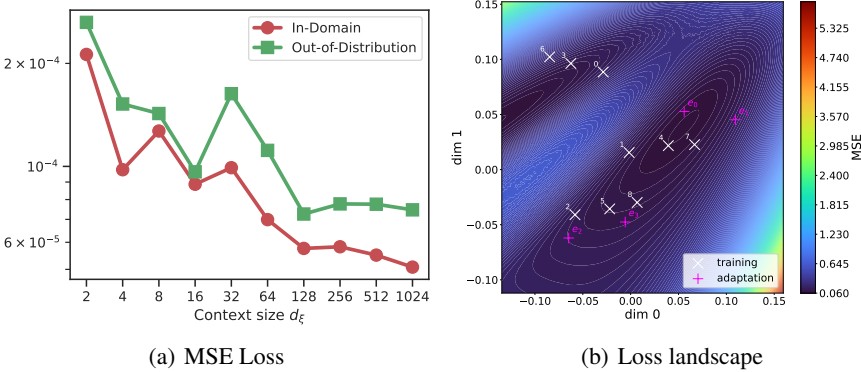

(a) MSE Loss  (b) Loss landscape

Figure 21: (a) InD and OoD evaluation MSEs for the LV problem as the context size is increased. (b) Loss landscape plotted in context vectors space for $d_\xi = 2$. The clustering of the training contexts mirrors the parameters pairs $(\beta, \delta)$ from Fig. 3.

### C.3 VARIATION OF THE POOLING STRATEGY

The context pool plays an important role in the NCF training process. In order to complement our comments in Appendix A.6, we test the effect of choosing various pooling strategies, namely Nearest-First (NF), Random-All (RA), and Smallest-First (SF). In this experiment, we focus on the LV case with $d_\xi = 2$ due to its linearity and our knowledge of the interpretable behavior of its contexts (see Appendix A.2). We train all three strategies with all hyperparameters identical

(including the neural network vector field initialization). We monitor the validation loss to select the best model across all 10000 epochs. The experiment is repeated 3 times with different seeds.

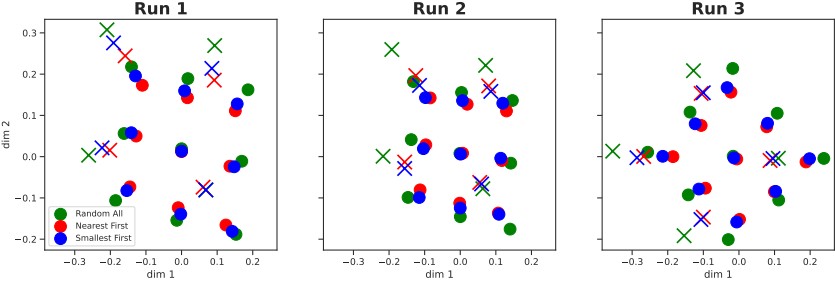

Figure 22: Both dimensions of the context vector post-training (indicated by dots), and post-adaptation (crosses) across several runs. Given the same neural network initialization, all pooling strategies are able to find the structure mimicking that of the underlying physical parameters observed in Fig. 5.

Although all strategies capture the underlying structure of the LV physical system (see Fig. 22), Table 6 indicates that RA and NF have the best performance, contrasting with high metrics and uncertainties displayed by SF. The training and validation dynamics in Fig. 23 provide clearer insight into this discrepancy, highlighting an important spread of validation values for SF. Interestingly, Fig. 23 shows that RA takes much longer to converge during training, despite ultimately producing the best OoD performance. This slow convergence is because very early on, the model is still forced to find commonalities between *all* pairs of environment, despite some pairs being less related than others. All in all, NF is the most balanced strategy on the LV problem since it converges fast, shows signs of increased stability, and produces excellent results InD and OoD.

Table 6: Comparison of pooling strategies on the LV problem. The reported mean and standard deviation training and sequential adaptation times are expressed in minutes. All strategies are trained for the same number of epochs across 3 separate runs with identical hyperparameters.

|  | RANDOM-ALL (RA) | NEAREST-FIRST (NF) | SMALLEST-FIRST (NF) |
| --- | --- | --- | --- |
| TRAIN TIME (mins) | $22.90 \pm 0.04$ | $22.82 \pm 0.03$ | $22.94 \pm 0.04$ |
| ADAPT TIME (mins) | $2.80 \pm 0.01$ | $2.80 \pm 0.01$ | $2.80 \pm 0.01$ |
| IND MSE ($\times 10^{-4}$) | $2.65 \pm 0.16$ | $2.52 \pm 0.66$ | $5.03 \pm 3.01$ |
| OOD MSE ($\times 10^{-4}$) | $2.38 \pm 0.43$ | $2.69 \pm 0.35$ | $3.34 \pm 0.96$ |

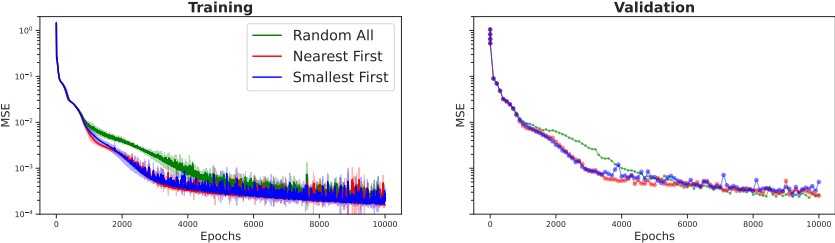

Figure 23: Loss curves with varying pooling strategies on the LV problem. (Left) Training; (Right) Mean validation losses across 3 runs, color-coded to match training curve labels.

## C.4 ABLATION OF THE 3-NETWORKS ARCHITECTURE

Another key element of the NCF framework is the 3-networks architecture described in Fig. 2b. In it, the vector field consists of state and context-specific networks that help bring the inputs into the same latent representational space before passing to another network to approximate the local derivative. Like the context size, its ablation directly contributes to a reduction in learnable parameter count.

Table 7: MSE upon ablation of the 3-networks architecture (NCF*) on both SP and LV problems, highlighting difficulties adapting to new environments.

| | SP ($\times 10^{-2}$) | | LV ($\times 10^{-5}$) | |
| | In-Domain | Adaptation | In-Domain | Adaptation |
|---|---|---|---|---|
| NCF | $11.0 \pm 0.3$ | $0.0003 \pm 0.00001$ | $6.73 \pm 0.87$ | $7.92 \pm 1.04$ |
| NCF* | $79.8 \pm 0.8$ | $1.6402 \pm 0.7539$ | $11.03 \pm 1.1$ | $69.98 \pm 84.36$ |

For this experiment, we consider NCF-$t_1$. We remove the data- and context-specific neural networks in the vector field, and we directly concatenate $x_i^e(t)$, and $\xi$; the result of which is passed to a (single) neural network. In this regard, NCF* (which designates NCF when its 3-networks architecture is removed) is analogous to CAVIA (Zintgraf et al., 2019). That said, NCF still benefits from the flow of contextual information, which allows for self-modulation.

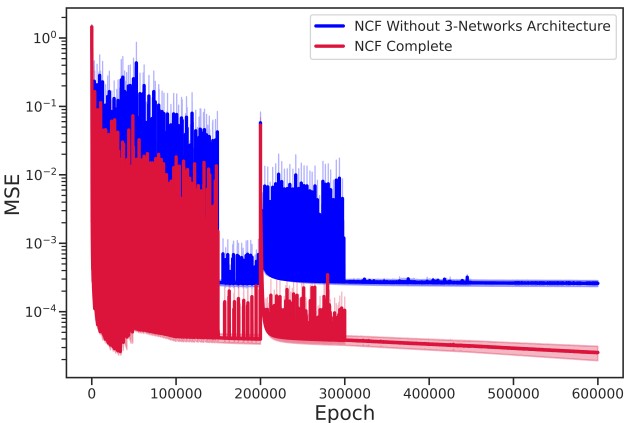

Figure 24: MSE loss curves when training the LV problem on a complete NCF, and on NCF*, the NCF variant deprived of the 3-networks architecture.

This study is of particular significance since, if in addition to foregoing the 3-networks architecture, we eliminated the Taylor expansion step, and NCF would turn into the data-controlled neural ODE (Massaroli et al., 2020). So we run both dynamics forecasting problems without the 3-networks architecture, and we report the training MSE in Fig. 24. In-domain and adaptation MSEs for both LV and SP problems are reported in Table 7.

While Fig. 24 highlights a performance discrepancy of nearly 1 order of magnitude during training, the key insight is hidden in the adaptation columns of Table 7. Indeed, the removal of the data- and context-specific networks in the vector field considerably restricts the model's ability to generalize to unseen environments, both for the SP and LV problems. We observed in our experiments, similar performance drops when only one of the two state or context networks was removed. This consolidates the 3-networks architecture as an essential piece of the NCF framework.

## D  EXPERIMENTAL DETAILS

This section shares crucial details that went into the experiments conducted in Section 4 and expanded upon in Appendices B and C. Unless otherwise specified, the hyperparameters in the next 3 paragraphs were applied to produce all figures and tables in this work.

Each method was tuned to provide an excellent balance of accuracy and efficiency. The main motivation behind our choices was to restrain model sizes both to allow efficient training of all baselines and fair comparison at roughly equal parameter counts. Irreconcilable differences with the baseline adaptation rules made it difficult to perform a systematic comparison, which is why we focused on parameter count as done in (Qin et al.); all the while noting that all models involved had sufficient capacity to approximate the problems at hand. For NCF and CAVIA, we counted the number of learnable parameters in the main network being modulated. For CoDA, we counted the root network plus the hypernetwork's learnable parameters as outlined in Eq. (3).

Our main workstation for training and adaptation was fitted with an Nvidia GeForce RTX 4080 graphics card which was used for the SP, LV, SM, and NS problems. Additionally, we used an RTX 3090 GPU of the same generation for the GO problem, and an NVIDIA A100 Tensor Core GPU for the GS problem as its CNN-based architecture was more memory-intensive. In addition to the training hardware and deep learning frameworks (JAX for NCFs, and PyTorch for CAVIA and CoDA) that made apple-to-apple comparison challenging, our network architectures varied slightly across methods, as the subsections below highlight. This is part of the reason we decided to focus on total parameter count as the great equalizer.

To ensure fair comparison, we made sure each model was sufficiently large to represent the task at hand (by comparing to commonly used hyperparameters in the literature, e.g., in (Kirchmeyer et al., 2022)). Since an "epoch" meant something different for the frameworks under comparison, we simply made sure the models were trained for sufficiently long, i.e., the lowest-performing method for each task was trained for at least as long as the second-best for that task, and its checkpoint with the best validation error was restored at the end.

### D.1  NCFs

**Model architecture**  The MLPs used as our context, state and main networks (as depicted in Fig. 2b) typically had a uniform width. When that was not the case, we use the notation $[h_{\text{in}} \to h_1 \to ... \to h_{d-1} \to h_{\text{out}}]$ to summarize the number of units in each layer of an MLP of depth $d$ for brevity. We used a similar notation to summarize the number of channels in a Convolutional Neural Network (CNN). For instance, on the **SP** problem, the context network was the MLP $[256 \to 64 \to 64 \to 64]$ (which indicates a context size of $d_\xi = 256$), the state network was $[2 \to 64 \to 64 \to 64]$, and the main network was $[128 \to 64 \to 64 \to 64 \to 2]$, all with Swish activations (Ramachandran et al., 2017). For the OFA and OPE experiments that do not require contextual information (cf. Tables 1 and 4), we delete the state and context networks then increase the hidden units of the main network to 156 to match the NCF parameter count. Below, we follow the same convention to detail the networks used for other problems in this paper's main comparison (see Table 2).

• **LV** (context) $[1024 \to 256 \to 256 \to 64]$; (state) $[2 \to 64 \to 64 \to 64]$; (main) $[128 \to 64 \to 64 \to 64 \to 2]$ • **GO** (context) $[256 \to 64 \to 64 \to 122]$; (state) $[7 \to 122 \to 122 \to 122]$; (main) $[244 \to 64 \to 64 \to 64 \to 7]$ • **SM** (context, state, main) Exact same architectures as with SP above • **BT** (context) linear layer with 256 input and 128 output features followed by an appropriate reshaping then a 2D convolution with 8 channels and circular padding to maintain image size, a kernel of size $3 \times 3$, and Swish activation[12]; (state) appropriate reshaping before 2D convolution with 8 channels; (main) CNN $[16 \to 64 \to 64 \to 64 \to 2]$ • **GS** Same network architectures as for the previous BT problem, with the exception that the linear layer in the context network outputted 2048 features • **NS** (context) A single linear layer with 202 input and 1024 output features reshaped and concatenated to the state and the grid coordinates before processing; (state) No state network was used for this problem; (main) 2-dimensional Fourier Neural Operator (FNO) (Li et al., 2020) with 4 spectral convolution layers, each with 8 frequency modes and hidden layers of width 10. Its lifting operator was a $1 \times 1$ convolution with 4 input and 10 output channels respectively – functionally

---

[12]Except for the number of channels, these were default convolution settings we used throughout. The circular padding enforced the periodic boundary condition observed in the trajectory data.

similar to a fully connected layer with weight sharing. The projection operator had two such layers, with 10 inputs, 1 output, and 16 hidden channels in-between.

**Training hyperparameters**    As for the propagation of gradients from the loss function to the learnable parameters in the right-hand-side of the neural ODEs (5), we opted to differentiate through our solvers, rather than numerically solving for adjoint states (Chen et al., 2018). Since our main solvers were the less demanding RK4 and Dopri5 solvers, we didn't incur the heavy memory cost Differentiable Programming methods are typically known for (Nzoyem et al., 2023; Kidger et al., 2020).

Specifically, we used the following integration schemes: Dopri5 (Wanner & Hairer, 1996) for GO, SM, GS, and BT, with relative and absolute tolerances of $10^{-3}$ and $10^{-6}$ respectively; RK4 for LV with $\Delta t = 0.1$; Explicit Euler for NS with $\Delta t = 1$. Across all NCF variants, we kept the context pool size under 4 to reduce computational workload ($p = 2$ for LV and SM, and $p = 4$ for LV and GO, and $p = 3$ for all PDE problems). We used the Adam optimizers (Diederik, 2014) for both model weights and contexts on ODE problems, and Adabelief (Zhuang et al., 2020) for PDE problems. Their initial learning rates were as follows: $3 \times 10^{-4}$ for LV and NS, $10^{-3}$ for GO, BT, and GS, $10^{-4}$ for SM. That learning rate was kept constant throughout the various trainings, except for BT, GS, and NS where it was multiplied by a factor (0.1, 0.5, and 0.1 respectively) after a third of the total number of training steps, and again by the same factor at two-thirds completion. The same initial learning rates were used during adaptation, with the number of iterations typically set to 1500.

As per Eq. (10), the NCF implementation was batched across both environments and trajectories for distributed and faster training. This was in contrast to the two baselines, in which predictions were only batched across trajectories. Using full batches to accelerate training meant that with LV for instance, all 4, 32, or 1 trajectories were used at once depending on the data split (meta-training vs. meta-testing, support vs. query). For regularization of the loss function Eq. (10), we set $\lambda_1 = 10^{-3}$ for all problems; but $\lambda_2 = 0$ for ODE problems and $\lambda_2 = 10^{-3}$ for PDE problems. For NCF-$t_2$ we always used a proximal coefficient $\beta = 10$ except for the LV where $\beta = 100$.

As for the simple pendulum SP problem used in this work, it leveraged NCF-$t_1$ with $p = 4$ and $d_\xi = 256$. The integrator was Dopri5 with default tolerances, a proximal coefficient $\beta = 10^2$, a constant learning rate of $10^{-4}$, and 12000 epochs. The OFA and OPE comparisons used the same hyperparameters, but with 6000 epochs for OPE and 2000 for OFA. For the other problems highlighted in Table 2, the remaining crucial hyperparameters (not stated above) are defined in Tables 8 and 9.

Table 8: Hyperparameters for NCF-$t_1$

| Hyperparameters | LV | GO | SM | BT | GS | NS |
|---|---|---|---|---|---|---|
| **Context size** $d_\xi$ | 1024 | 256 | 256 | 256 | 256 | 202 |
| **Pool-filling strategy** | NF | RA | RA | NF | RA | NF |
| **# Epochs** | 10000 | 2400 | 24000 | 10000 | 10000 | 5000 |

Table 9: Hyperparameters for NCF-$t_2$

| Hyperparameters | LV | GO | SM | BT | GS | NS |
|---|---|---|---|---|---|---|
| **Context size** $d_\xi$ | 1024 | 256 | 256 | 256 | 256 | 202 |
| **Pool-filling strategy** | NF | NF | RA | NF | RA | NF |
| **# Inner iterations**[13] | 25 | 10 | 10 | 20 | 20 | 25 |
| **# Outer iterations** | 250 | 1000 | 1500 | 1000 | 700 | 250 |

---

[13]The inner iterations used in NCF-$t_2$ are not to be confused with the inner gradient update steps in CAVIA, since NCF performs alternating rather than bi-level optimization.

## D.2    CoDA

The reference CoDA implementation from (Kirchmeyer et al., 2022) was readily usable for most problems. We adapted its data generation process to incorporate two new benchmarks introduced in this paper (the SM and BT problems).

**Model architecture**    Neural networks without physics priors were employed throughout, all with the Swish activation function. • **LV**: 4-layers MLP with 224 neurons per hidden layers (width) • **GO**: 4-layers MLP with width 146 • **SM**: 4-layers MLP with width 90 • **BT**: 4-layers ConvNet with 46 hidden convolutional filters and $3 \times 3$ kernels. Its output was rescaled by $10^{-4}$ to stabilize rollouts • **GS**: Same ConvNet as BT, but with 106 filters. • **NS**: 2-dimensional Fourier Neural Operator (FNO) (Li et al., 2020) with 4 spectral convolution layers, each with 8 frequency modes and hidden layers of width 10. Its lifting operator was a single fully connected layer, while its projection operator had two such layers, with 16 hidden neurons.

**Training hyperparameters**    Using `TorchDiffEq` (Chen, 2018), CoDA backpropagates gradients through the numerical integrator. We used the same integration schemes as NCF above. We stabilized its training by applying exponential Scheduled Sampling (Bengio et al., 2015) with constant $k = 0.99$ and initial gain $\epsilon_0 = 0.99$ updated every 10 epochs (except for the **NS** problem updated every 15 epochs). The Adam optimizer with a constant learning rate of $10^{-4}$ was used. The batch size and the number of epochs are given in the Table 10.

Table 10: Hyperparameters for CoDA

| Hyperparameters | LV | GO | SM | BT | GS | NS |
|---|---|---|---|---|---|---|
| **Context size** $d_\xi$ | 2 | 2 | 2 | 2 | 2 | 2 |
| **Minibatch size** | 4 | 32 | 4 | 1 | 1 | 16 |
| **# Epochs** | 40000 | 40000 | 12000 | 10000 | 120000 | 30000 |

We note that in line with CoDA's fundamental low-rank assumption, we maintained a context size $d_\xi = 2$ throughout this work since no more than 2 parameters varied for any given problem.

## D.3    CAVIA

The reference implementation of CAVIA-Concat (Zintgraf et al., 2019) required substantial modifications to fit dynamical systems. Importantly, we incorporated the `TorchDiffEq` (Chen, 2018) open-source package and we adjusted other hyperparameters accordingly to match NCF and CoDA on parameter count and other relevant aspects for fair comparison. The performance of CAVIA-Concat was heavily dependent on the context size, which we adjusted depending on the problem.

**Model architecture**    Like CoDA, the Swish activation was used for all neural networks • **LV**: 4-layers MLP with width 278 • **GO**: 4-layers MLP with width 168 • **SM**: 4-layers MLP with width 84 • **BT**: 7-layers ConvNet with 46 hidden convolutional filters and $3 \times 3$ kernels. Its outputs was rescaled by $0.1$ to stabilize rollouts • **GS**: 4-layers ConvNet with 184 hidden convolutional filters and $3 \times 3$ kernels. Its outputs was rescaled by $0.1$ to stabilize rollouts. • **NS**: Same FNO as with NCF above, including the single-layer context network with 1024 output neurons.

**Training hyperparameters**    We used full batches to accelerate training. We use the Adam optimizer with a learning rate of 0.001 for the meta-update step. The single inner update step had a learning rate of 0.1. The number of iterations is given in Table 11.

Table 11: Hyperparameters for CAVIA

| Hyperparameters | LV | GO | SM | BT | GS | NS |
|---|---|---|---|---|---|---|
| **Context size** $d_\xi$ | 1024 | 256 | 256 | 64 | 1024 | 202 |
| **# Iterations** | 200 | 100 | 1200 | 500 | 50 | 1200 |

## E    EXAMPLE IMPLEMENTATION OF NCFS

A highly performant JAX implementation (Bradbury et al., 2018) of our algorithms is available at `https://github.com/ddrous/ncflow` [14]. We provide below a few central pieces of our codebase using the ever-growing JAX ecosystem, in particular Optax (DeepMind et al., 2020) for optimization, and Equinox (Kidger & Garcia, 2021) for neural network definition.

**Vector field**    The vector field takes center-stage when modeling using ODEs. It is critical to use JVPs, since we never want to materialize the Jacobian (the context size can be prohibitively large). The Jacobian-Vector Product primitive from Equinox (Kidger & Garcia, 2021) `filter_jvp` as illustrated below, coupled with `jit`-compilation, enabled fast runtimes for all problems.

```python
class ContextFlowVectorField(eqx.Module):
    physics: eqx.Module
    augmentation: eqx.Module

    def __init__(self, augmentation, physics=None):
        self.augmentation = augmentation
        self.physics = physics

    def __call__(self, t, x, ctxs):
        ctx, ctx_ = ctxs
        # ctx = \xi^e, while ctx_ = \xi^j

        if self.physics is None:
            vf = lambda xi: self.augmentation(t, x, xi)
        else:
            vf = lambda xi: self.physics(t, x, xi) + self.augmentation(t, x, xi)

        gradvf = lambda xi_: eqx.filter_jvp(vf, (xi_,), (ctx-xi_,))[1]
        scd_order_term = eqx.filter_jvp(gradvf, (ctx_,), (ctx-ctx_,))[1]

        return vf(ctx_) + 1.5*gradvf(ctx_) + 0.5*scd_order_term
```
Listing 1: Second-order Taylor expansion in the NCF vector field

**Loss function**    Eq. (10) provides a structured summation layout particularly suited for a function transformation like JAX's `vmap`. During implementation, that loss functions can be defined in two stages, the innermost summation term along with Eq. (9) making up the first stage; while the two outermost summations of equation 10 make up the second stage. Once the first stage is complete, the second is relatively easy to implement. We provide a vectorized implementation of the first stage below.

```python
def loss_fn_ctx(model, trajs, t_eval, ctx, all_ctx_s, key):
    """ Inner loss function Eq. 9 """

    ind = jax.random.permutation(key, all_ctx_s.shape[0])[:context_pool_size]
    ctx_s = all_ctx_s[ind, :]                    # construction of the context pool P

    batched_model = jax.vmap(model, in_axes=(None, None, None, 0))

    trajs_hat, nb_steps = batched_model(trajs[:, 0, :], t_eval, ctx, ctx_s)
    new_trajs = jnp.broadcast_to(trajs, trajs_hat.shape)

    term1 = jnp.mean((new_trajs-trajs_hat)**2)  # reconstruction error

    term2 = jnp.mean(jnp.abs(ctx))              # context regularisation

    term3 = params_norm_squared(model)          # weights regularisation

    loss_val = term1 + 1e-3*term2 + 1e-3*term3

    return loss_val
```
Listing 2: Inner NCF loss function with vectorization support

---

[14] A PyTorch codebase is equally made available at `https://github.com/ddrous/ncflow-torch`.

# F    TRAJECTORIES VISUALIZATION

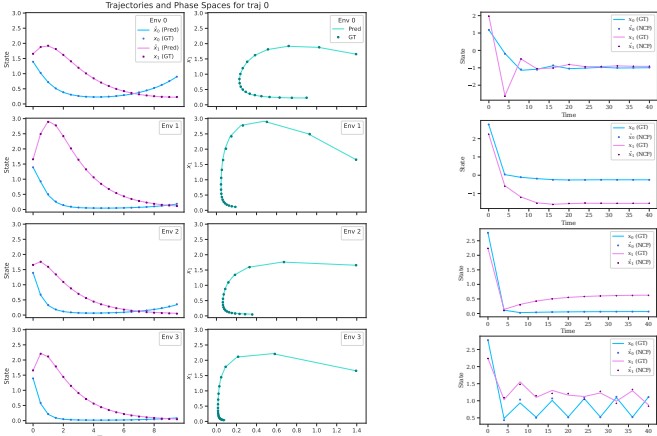

Figure 25: **(Right)** Visualizing the first ground truth and predicted testing trajectories and phase spaces in all 4 adaptation environments of the LV problem. The initial condition is the same across all 4 adaptation environments. **(Left)** Visualizing the first ground truth testing trajectory in 4 meta-training environments found in various attractors of the SM problem: the first in (L1), the second and third in (E), and the fourth in (L2).

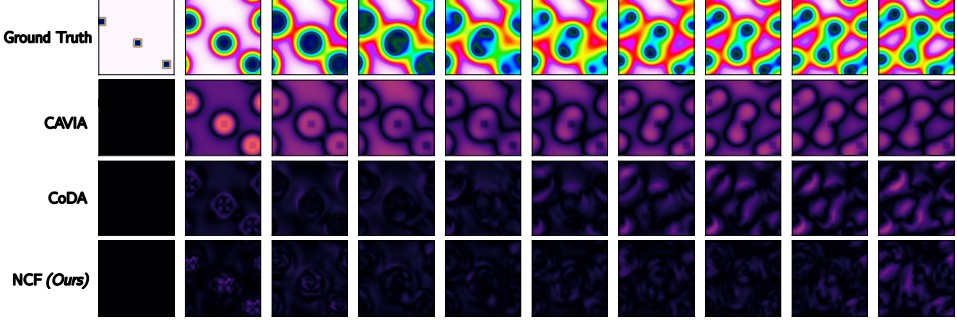

Figure 26: Sample absolute reconstruction error of a trajectory during adaptation of the Gray-Scott system with NCF-$t_2$. Trajectories begin at $t = 0$ (left) and end at $t = 400$. This describes the sole trajectory in the fourth adaptation environment.

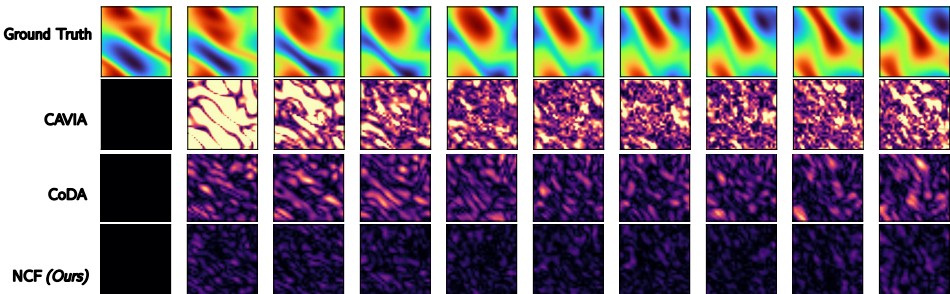

Figure 27: Sample absolute reconstruction error of a trajectory during adaptation of the Navier-Stokes system with NCF-$t_2$. Trajectories begin at $t = 0$ (left) and end at $t = 10$. The viscosity for the reported environment is $\nu = 1.15 \times 10^{-3}$.

