# OpenReview forum: "Neural Context Flows for Meta-Learning of Dynamical Systems"
_ICLR.cc/2025/Conference — ICLR 2025 Poster_

### Official Review · Reviewer_fPSV · 2024-10-20

**Soundness:** 3
**Presentation:** 3
**Contribution:** 2
**Rating:** 6
**Confidence:** 3

**Summary:**

This paper proposes NCF to solve parametric ordinary differential equations. It is designed to improve the adaptability and generalization of learning dynamical systems across various environments. NCF introduces a meta-learning approach that employs a context-modulation mechanism, incorporating uncertainty estimation. Specificaly,  NCF uses k-th order Taylor expansion to enable contextual self-modulation. The authors demonstrate the performance of NCF on a variety of ODEs and PDEs problems and illustrate the effectiveness of the proposed methods.

**Strengths:**

(1) Introducing a meta-learning framework to study complex dynamical systems driven by ODEs and PDEs is a novel and interesting direction.
(2) This article includes both experimental validation and some theoretical analysis.

**Weaknesses:**

(1) Currently, there are several methods for parameterizing equations, such as those for ordinary differential equations [1] and partial differential equations [2]. In the paper, the authors mainly compared their approach with meta-learning methods like CAVIA and CODA. So, does it have a competitive advantage over other non-meta-learning methods?
[1] Parameterized Neural Ordinary Differential Equations: Applications to Computational Physics Problems
[2] Identification of the flux function of nonlinear conservation laws with variable parameters

(2) The authors mentioned that their method is robust. One important aspect of proving robustness is how the model performs when the observed data contains noise. However, this was not demonstrated in the experiments.

(3) The authors converted three PDE problems into ODE problems for their study. They should provide a detailed description of how this conversion was done and how the errors were controlled during the process. PDE problems are quite sensitive to the choice of numerical schemes, and different discretization methods can significantly affect the accuracy of the solution. Converting them into ODEs and solving them using an ODE solver will inevitably introduce errors.

(4) The link to the code provided in the paper is inaccessible.

**Questions:**

Refer to weakness.

---

> ### Author Response · Authors · 2024-11-23
>
> Thank you for reading our paper, and for noticing its novel contribution and direction. We are very pleased you found that we included both experimental validation and theoretical analyses. You might be pleased to learn that we've added __Proposition 2__ for identifiability of affine systems.
>
> Below, we address the concerns that were raised.
>
> ---
>
> ### W1. What is the competitive advantage over non-meta learning methods ?
> Yes, our meta-learning approach offers competitive advantage over non-meta learning baselines. We acknowledge that in the original paper, we did not do enough to show this. To that end, we've implemented the __One-Per-Env__ (one _context-free_ model trained for _each_ individual environment) from [1] and conducted a full analysis comparing it to __One-For-All__ (one _context-free_ model for _all_ environments at once), and to our NCF (one _context-informed_ model for _all_ environments at once). Our results are reported in __Appendix C.3__ starting at line 1337, and they show that our approach is beneficial in low-data regimes as it offers low MSEs and low amortized training times.
>
> We thank the reviewer for the two references which we found particularly relevant. We've added them to the Introduction in the paragraph motivating Neural ODEs.
>
> [1] Yin et al. LEADS: Learning dynamical systems that generalize across environments, NeurIPS, 2021.
>
> ---
>
> ### W2. Robustness to noise
> We appreciate this remark by the reviewer. Indeed, in our original submission, we did not do enough to clarify how our model is robust. In the updated manuscript, we have demonstrated robustness to noise experimentally in __Appendix A.2__ (around __line 900__) which shows how well our method performs when increasing amounts of Gaussian noise are added to the single adaptation trajectory. More importantly, we have removed the term "robust" in our contributions since at that point early in the paper, it is not clear what robustness means. Elsewhere in the paper, we believe it should be clear that by robustness, we refer to noise in the adaptation trajectory.
>
> ---
>
> ### W3. Trade-offs when converting PDEs to ODEs.
>
> We agree with the reviewer that converting our PDEs into ODEs via the method of lines has several drawbacks. In our response below, we focus mainly on errrors due to the spatial and temporal discretization, and the one related to boundary conditions.
>
> During the data-generation process, the spatial discretization of the PDE grids were all coarse, with a uniform cell spacing of $\Delta s=1$. Because of that, step size error was locally controlled by using the RK4 adaptive time-step initial value solver implemented in `solve_ivp` from Scipy, just like in [1,2]. This ensured that the integrator always took small enough time steps for the CFL condition of the PDE at hand to be satisfied, thus avoiding instabilities.
>
> During training, however, we found that simply using differentiable adaptive time-steppers was not enough to avoid blow-ups of rollouts. For that reason, we multiplied the output of the neural network vector field by a scale of $10^{-2}$. This is a strategy we found in [2] (as we stated in Appendix B.2, __line 1143__), which equally worked well for our NCF implementation and for CAVIA [3].
>
> Concerning the error at the domain boundaries, they were mitigated by using __periodic__ boundary conditions. We inherited the data-generation process of [2] which was easily replicated with NumPy. The main point is that only considering PDEs with periodic boundary conditions in our work ensured that convolutional layers with __circular padding__ could be readily used in the neural network field for effective modelling.
>
> [1] Yin et al. LEADS: Learning dynamical systems that generalize across environments, NeurIPS, 2021.
>
> [2] Kirchmeyer et al., Generalizing to New Physical Systems via Context-Informed Dynamics Model, ICML, 2022.
>
> [3] Zintgraf et al. Fast Context Adaptation via Meta-Learning, ICML 2019.
>
> ---
>
> ### W4. Inaccessible links
> We apologize for this. The URL links were set as placeholders to protect our anonymity. We submitted both our code and our Gen-Dynamics datasets in archive format. We have now made an [anonymous repository](https://anonymous.4open.science/r/neural-context-flow/README.md), and the links to our code in the revised PDF should point to it. The same goes for the [Gen-Dynamics](https://anonymous.4open.science/r/gen-dynamics/) initiative.
>
> ---
>
> We once again wish to thank you for your great reviews, for your positive evaluation and support. We hope we have addressed your concerns, and we hope to continue in a great discussion if some concerns were unaddressed.

---

> ### Author Response · Authors · 2024-11-30
> **A gentle reminder for rebuttal acknowledgment**
>
> Dear Reviewer,
>
> Thank you once more for reviewing our work and for providing such insightful feedback. We have carefully responded to all your comments and questions; and our manuscript was appropriately updated.
>
> The new deadline for the author-reviewer discussion is coming up soon. Could you please take a moment to review our responses ? If you find them satisfactory, we hope you might consider adjusting your initial rating.
>
> Any additional comments or feedback would be equally welcome and swiftly acted upon.
>
> Thanks a lot
>
> Authors

---

### Official Review · Reviewer_6smX · 2024-10-28

**Soundness:** 2
**Presentation:** 2
**Contribution:** 2
**Rating:** 5
**Confidence:** 3

**Summary:**

The paper introduces a new method for meta-learning of dynamical systems by enforcing context vector to be smooth and close to each other using a training method based on Taylor expansion of the vector field.

**Strengths:**

The paper introduces a method that seems reasonable, with several interesting analysis of the behavior of the learned vector field. The related works section does present the most popular baselines and methods for this problem. The datasets used to evaluate the method are in par with what is currently used in the litterature.

**Weaknesses:**

I have several major concerns on this paper that I will try to rank from higher to lower importance.
1) **Overfit**: I am very confused by the OoD adaption protocol used in the paper. From what I understood, Algorithm 2 is used on the validation set to tune the value of $\xi$ using gradient descent. Then, the prediction error is computed **on the same trajectories as the one used to pick $\xi$**. Given the size of $\xi$, it is most certainly overfitting to the (small) set of trajectories used to tune the context vector. I suspect that the value of $\xi$ is specifically tuned to match a small set of trajectories during validation, hence the results are not representative of the true performance of the model. A quick look at the code seems to confirm that, although it is hardly readable (many commented functions, no comments, empty README, residual files from project development). Moreover, the model performs almost systematically better in OoD setup than in In-domain, (table 1 and 2), which is unexpected, and very unusual.
2) **Unsupported claims**:
	- "we introduce a *robust*  ..." ? The word "robust" is never used in the experimental section.
	- line 248 : "*The $L^1$ regularization term is particularly vital in promoting sparsity in possibly high dimensional context vector*". Sparsity is indeed used a lot for equation retrieval, such as SinDy-like methods, but I don't see why this is crucial here. Yet it probably is, but I don't see which experiment in the paper justify this.
	- "*straightforward method for uncertainty quantification*" : seems like an overstatement according to my understanding of the experiment. Quantifying the uncertainty means that the model can provides an interval to which the true trajectory must belong (up to some probability score). From what I understand from your experiment, you consider that the predictions from your model obtained with different contexts (including unrelated to the current environment) gives an uncertainty on the prediction.
1) **Motivation of the method**: the use of Taylor development is motivated (section 3, l.200 to 206) by the fact that it forces context vector to remain close. It is not clear to me why closeness is important for the task, and more importantly why simpler regularization to force $\xi_i$ to remain close from each other would not perform well.
2) **Complexity of the method**: training NCF requires to compute the second order derivative of a neural network, and then back-propagate through the entire computation graph, including the use of a Neural ODE. The method seems utterly complex and computationally demanding. Moreover, some design choices are unclear to me (see questions). Many hyper-parameters have to be set, and the paper provides no clear indication on how to set them (mostly, size of the context vector and context pool mode (RA, NF or SF)). However, I did appreciated the experiments on the size of the context pool.

	Finally, Figure 7 and 16 shows training curves of the model exhibiting high spikes, including one (figure 7) from which the model never recover. This seems to indicate a highly unstable and difficult training, which is not a good sign for extrapolation to more complex dynamics.

1) **Experimental section**: It seems that the order 1 NCF model is trained with a different algorithm than the order 2 (l.264-268). I am not sure to see why, and more importantly, if the gap between the two models in table 1 is due to the supplementary order of the Taylor expansion, or the different training algorithm.

**Minor** (no effect on rating)
- in the introduction: "*Its dynamics are heavily depending on its parameter*" looks like an overstatement. The dependance of a dynamical system on its parameter can vary significantly from a system to another.
-  In section 2, you mentioned that CoDA involves two networks instead of one, hence requiring more computational resources to train. Your approach uses three networks. You might clarify this statement to explicitly mentioned hyper-networks as the bottleneck, and not the use of two networks ?
- in the beginning of section 3 (l.168), $\mathcal{D}_{ad}$ is mentioned before being introduced.
- L. 171, you refer to the smoothness assumption as an inductive bias. It's closer to a constraint than a real inductive bias (or at least, it is an inductive bias on the type of dataset you will test your model on, but not an inductive bias of general physical systems).

**Motivation for my grade**
My grade is mostly driven by my suspicion of overfitting. However, I am also concerned by the (in my opinion) poorly motivated design choices (use of Taylor expansion to promote closeness of context vector, supplementary encoder for context vector, back-propagating through the Hessian) and my difficulty to understand several claims and experiments.

**Questions:**

1) Did you used **different** trajectories to compute the metrics in the tables than the one used to perform the adaptation in OoD setup?
2) Could you please explain why your model performs better in OoD than InD ? This is an unexpected resuls : we would expect the model to perform always better on seen data.
3) Could you clarify what *robust* means in this context ? (robustness to OoD, to noise, to unseen initial condition ? ) and point to the corresponding experiment that support the claim ?
4) Could you elaborate on the motivation of using Taylor expansion for training the model ? If the main reason behind this is to force context to remain close, then a crucial ablation is missing where the Taylor expansion is replaced by a simple distance loss between context vector.
5) The context vector is a learned embedding, so what is the point of learning a supplementary dedicated network to convert it into $\tilde \xi$, since you could directly learn $\tilde \xi$ ? In the paper, you explains that "this allows the framework to automatically balance the potentially non linear influence of the context with that of the state vector". What does this means ? Could you please elaborate on this ?
6) Could you please justify the claim line 248 that $L^1$ regularization is crucial ? Did I miss the ablation of this regularization in the paper ?
7) I am not sure to understand fig. 6. It seems that you collected the prediction of NCF for a same initial condition with different context vectors (including unrelated ones) and consider the max/min of these prediction as uncertainty measure. Hence my questions:
	- What is the probability of the true solution to lie within these bounds ?
	- It seems that all model gives fairly similar outputs, which is surprising since they are adapted to different dynamics. How do you explain that ?

**Minor**
 - I do appreciate code release. May I ask you to document and clean the code before publication ?
 - Could you please fix the url link to the associated github repo in the paper ?
 - Could clarify if Theorem 3.2 should be considered as a contribution ? It seems to be a straightforward application of Theorem 2 in Li et al, 2019. If not, it might be interesting to provide more details about what changed, and for which reason.

# Rebuttal update
The authors have provided a thorough reply to most of my concerns and have clearly demonstrated that my suspicion of overfitting was unfounded. This significantly impacts my evaluation, and I have raised my score from 1 to 5.

However, I still recommend rejecting this paper. While the proposed method is interesting and well-supported, it would benefit from further development, particularly in the analysis of its behavior. Although the authors made commendable efforts to improve their manuscript, the short timeframe allocated for rebuttals limited the depth of the supplementary results included. Specifically:

- The emergence of clusters of "similar" environments due to the context pool-filling method, coupled with the Taylor expansion constraint, is a compelling idea. However, it warrants deeper exploration, perhaps with more environments and a detailed analysis of the content of these clusters. These behavior is at the heart of the success of NCF and deserve a better, deeper exploration.
- Some design choices remain unclear, particularly the use of a projection network on the context embeddings.
- The authors have strengthened the "uncertainty"-related experiments, but this discussion should have been made clearer in the main paper. However, addressing this would require significant restructuring of the paper, which is impractical within the rebuttal timeline.

For these reasons, I have updated my evaluation to a borderline rejection. While the paper presents interesting results, I believe it is not yet ready for acceptance.

---

> ### Author Response · Authors · 2024-11-23
>
> Dear reviewer, we thank you for your time to read our work and its several interesting analyses. We appreciate that you find the Related Work section and the datasets important and representative of the state of the literature in this field.
>
> We've sought to address your concerns in the same order they were issued, from higher to lower importance. We merged minors weaknesses and questions to address them in small self-contained passages.
>
> ---
>
> ### Q1. Overfitting - Did we use __different__ trajectories to adapt and evaluate new $\xi$ ?
> Yes. We always adapt $\xi$ on a single trajectory, but then we sample entirely new ones for evaluation (albeit from the same initial condition distribution). We've actually gone to great lengths to select new trajectories, repeating the same exact ones from problems in CoDA [1]. To improve reproducibility, we've released the datasets as our third contribution, the Gen-Dynamics initiative (see __Appendix C.1__), whose `ood_test` split corresponds to adaptation-time evaluation samples. We have now re-uploaded our code to an [anonymous repository](https://anonymous.4open.science/r/neural-context-flow/README.md), with the main script properly commented, hopefully clarifying that adaptation training and testing samples are different. Finally, to avoid such confusions about overfitting, we've reiterated in line 407 of the revised PDF's main text that 32 adaptation testing trajectories of the SP problem are different from the one used to fine-tune the context, as was previously stated in __Section 3.4__.
>
> [1] Kirchmeyer et al., Generalizing to New Physical Systems via Context-Informed Dynamics Model, ICML, 2022.
>
> ---
>
> ### Q2. Better OoD performance
> We agree that better OoD performance is not intuitive in the general Machine Learning literature. However, our problem requires aggregating the MSE across various environments. Not all environments are equally well-resolved, as we see in __Figure 4c__ for the SM problem for instance. Now, since we use __mean__ metrics to aggregate these losses, the results in Table 1 will depend on how well the InD environments were resolved, and how close the OoD environments are to them (See __Fig 3__ for LV for instance). So __this is due to the datasets and the metrics__, not the methods. In fact, the same observation regarding better OoD performance can be observed in [1].
>
> [1] Kirchmeyer et al., Generalizing to New Physical Systems via Context-Informed Dynamics Model, ICML, 2022.
>
> ---
>
> ### Q3. What does _robust_ mean in this context ?
> We mean __robustness to noise__ as experimentally evidenced in the revised PDF's __Appendix A.2__ (around line 900) which shows how well our method performs when noise is added to the single adaptation trajectory (also line 492). In line with the reviewer's comments, we've also removed the term "robust" from our contributions passages, since it's not clear at that point what robustness means.
>
> ---
>
> ### Q4. Motivation for using Taylor expansion for training the model
> We thank you for this question. To complement our intuitive explanation around line 200, we add that the closeness of context vectors reflects our knowledge that the underlying dynamics are expected to be close to each other as well. But, importantly, we don't know how close, whether those are clustered, or whether outlier environments exit. This stems from the fact that the underlying parameters are typically unobserved, as we motivate in the Introduction.
>
> For instance, assuming only the nearest contexts are used in the pool, our method should _automatically_ encourage those contexts that are most related to stay together (for instance, __Figure 15__ in Appendix C.5), and repel others so they can form their own clusters. We do not want __all__ contexts to be equally close to each other. We want a form of proximity that reflects that of the physical parameters.
>
> Furthermore, building more on mathematical intuition, if we know that the vector field we want to approximate is differentiable wrt to its parameters, we would want the neural network to be differentiable wrt contexts as well. This is a constraint Taylor expansion enforces implicitly, and our __Proposition 2__ demonstrates that this results in a __provably identifiable affine system__ (something we are not sure we can get if we use simpler forms of regularization). We note that Proposition 2 was only added to the revised PDF, although previously stated informally in Appendix A.2.
>
> ---

---

> ### Author Response · Authors · 2024-11-23
>
> ### Q5. What is the point of a context network ?
>
> If we were to directly concatenate $x$ and $\xi$ before feeding into the main network, then its first Linear layer would represent the sum of two linear transformations of $x$ and $\xi$. (This is because concatenation-based conditioning is equivalent to additive conditioning, assuming some conditions are met [2]). This __sum of linear representations__ would then flow into subsequent layers on the neural network. That said, the relation between $x$ and the underlying parameter $c$ we wish to model might be a __non-linear__ one. This is why the state and context networks are used to form $\tilde x$ and $\tilde \xi$ in the hope that those two can interact linearly. We believe this 3-networks architecture adds expressivity to the model, without increasing the total parameter count, as __Table 1__ can attest.
>
> We agree with the reviewer that the mentioned sentence is difficult to parse, so we rephrased it in the revised PDF for added clarity (__line 212__).
>
> [2] Dumoulin, et al., "Feature-wise transformations", Distill, 2018.
>
> ---
>
> ### Q6. On the importance of L1 regularization
> We agree with the reviewer that line 248 was poorly phrased, and excessively emphasized the importance of L1 regularization. To clarify, L1 regularization is __neither vital nor crucial__ in our case, even though it helps in setting up Theorem 1 with __line 935__ (previously Theorem 3.1). To avoid possible misunderstandings, we've removed that sentence from the revised manuscript.
>
> ---
>
> ### Q7. Uncertainty quantification
> Indeed, the reviewer is correct in their understanding that candidate predictions were collected and used. However, in __Figure 6__, the shaded regions do not correspond to the max/min predictions, but rather to the scaled standard deviations across these candidates. In the manuscript, this is made clear in line 501.
>
> To fully address the reviewer's concerns, we conducted a full experiment on uncertainty estimation with NCF, the results are presented in __Appendix C.2__. They include the __Confidence Level__ which accounts for the probability of the true solution to lie within the bounds of a confidence interval.
>
> Candidates look similar because each individual adaptation was good enough and as a result, all contexts are close together. We should emphasize that when a neighboring context $\xi^j$ is used for prediction of a trajectory in $e$, $\xi^e$ is still used as per __Eq 6__. (As a consequence of first-order Taylor expansion for instance, the residual error is directly proportional to the difference between $\xi^e$ and $\xi^j$. The closer they are, the lower the approximation error). Therefore, in __line 1243__ of the revised manuscript, we've emphasized that unrelated OoD contexts should only be used for uncertainty estimation if the model performs well in those OoD regions.
>
> ---
>
> ### Weakness 4. Perceived complexity of the method
> We appreciate the feedback regarding the perceived complexity of our method. We've made efforts to expose the methods as intuitively as possible, and we respectfully disagree with the reviewer on some comments as follows:
> - The second-order derivatives, if needed with NCF-t2, are computed wrt __the contexts__ (which are relatively low-dimensional). Since we use alternating minimization, the neural network __weights are fixed when $\xi$s are optimized__. Furthermore, our proposition on JVPs avoids high costs as __we never compute nor do we back-propagate through the Hessian__. Our paper contains a section dedicated to scalability and computational demands in __Appendix A.6__
> - We do not see how using Neural ODEs makes our method more complex. As we motivate in our revised introduction, they form the backbone of so many parametric PDE solving frameworks (including the baselines in this paper) due to their flexibility and easy of use.
> - We have provided an analysis in the updated manuscript __Appendix D.3__ to shed some light on the context pooling strategy. The added hyperparameters are indeed a limitation, which we have acknowledge towards the end of our main text, and will be mitigating in future work.
> - The __spikes__ in the figures correspond to our reduction of the learning rate during training, as we mention in the Appendix B. We have replaced Figure 7 with the smoother __Figure 10__ in the revised manuscript.
>
> ---
>
> ### Weakness 5. Experimental section
> The mentioned gap is primarily observed on non-linear problems, where __NCF-t2__ is expected to outperform __NCF-t1__. In practice, we observed good results when using the ordinary alternating minimization algorithm throughout, but needed the proximal algorithm to obtained SoTA results. We have clarified this in the discussion around __Table 1__ in the revised PDF.
>
> ---

---

> ### Author Response · Authors · 2024-11-23
>
> ### Minor Questions
> - Yes, we agree that our original code was not particularly well-documented. Its README was emptied to preserve our anonymity in case a reviewer performed a GitHub search. Our new anonymous repository has a main script that is well documented. The entire codebase with library files will be cleaned and documented appropriately before publication.
> - The URL was simply a placeholder. It now points to the aforementioned [repository](https://anonymous.4open.science/r/neural-context-flow/README.md).
> - Our work mainly proposes 3 contributions, and we do not count Theorem 3.1 (renumbered simply as __Theorem 1__) as one of them. Indeed, it is a simple application of Li et al. 2019. We note that on the theoretical side, __Proposition 2__ was added to compensate for the theory limitations we mentioned in the discussion.
>
> ---
>
> ### Minor weaknesses (no effect on rating)
> That you for pointing out these minor issues.
> - We've rephrased the sentence in line 032 to "Its dynamics are influenced by its parameters"
> - We've rephrased the last paragraph on Meta-Learning to clearly highlight the limitations of CoDA and hypernetworks.
> - To void notation overload, we've now indicated that $D_{ad}$ is defined in a similar way as $D_{tr}$
> - We've replaced "inductive bias" with the more specific term "constraint" as appropriate.
>
> ---
>
> Again, thank you, your questions have helped use improve our manuscript. We believe to have addresses all concerns, __especially those that motivated your grade__. If the event that we missed a concern, we have happy to help clarify them until they are fully addressed.

---

> ### Comment · Reviewer_6smX · 2024-11-26
> **Reply**
>
> Thank you for your thorough reply, and your work and reactivity to answer the comments of the reviewers.
>
> **Q1**
> Thank you for this clarification. This is reassuring. I also greatly appreciate the effort you made to release the code, which significantly improves readability.
>
> **Q2**
> Thank you for your response, but I am not entirely convinced by your explanation. My understanding of your argument is that simpler environments indeed give lower mean prediction scores, and smoother dynamics with respect to the parameters make out-of-distribution (OOD) adaptation easier, thereby reducing the mean score. However, I fail to see how this phenomenon would explain why OOD scores are lower than in-distribution (ID) scores, as these two aspects do not seem connected. Simpler and smoother dynamics might result in lower errors for ID and enable easier adaptation, but OOD samples remain fundamentally OOD—exploring regions unseen during training, where performance generally deteriorates.
>
> That said, ruling out overfitting as the cause makes this issue less concerning. However, I am now skeptical about the complexity of certain tasks, as some dynamics seem simple enough to allow adaptation to the point where prediction scores improve for OOD cases.
>
> **Q3**
> Thank you for updating your manuscript. NCF indeed demonstrates strong robustness against noise, particularly compared to CODA.
>
> **Q4**
>
> Thank you for your explanation, which helps clarify your contribution. The Taylor expansion approach makes sense, especially when paired with a dynamic context pool-filling method. I find the emergence of local “clusters” from your training approach quite interesting. However, the t-SNE visualization in Figure 15 is unconvincing due to the small number of points.
>
> In my opinion, this study would benefit from a more detailed analysis, including more points, insights into how these clusters emerge, what they contain, and how the model performs within each cluster. This aspect, in my view, is central to the method.
>
> However, I understand that such an analysis goes beyond the scope of a rebuttal and would require training on additional environments, which may conflict with the "data scarcity" constraint outlined in the introduction.
>
> **Q5**
>
> Perhaps my original question was unclear; allow me to rephrase it. The parameter $\xi$ is a learnable embedding associated with one environment. This parameter is unconstrained in its structure (except via the loss) and can therefore take any value in $\mathbb{R}^{d_\xi}$.
>
> You apply a nonlinear transformation to $\xi$ to obtain $\tilde{\xi}$ before feeding it into the dynamics. My question is: what prevents gradient descent from directly identifying $\tilde{\xi}$?
>
> I can see a potential motivation for this additional network by referring to the linear probing experiment, where you recover the true physical parameters from the embedding using a simple linear layer. For the linear probe to perform well, there is likely a need for nonlinearity before concatenation with the state in the main model. However, justifying a design choice in the main model based on the success of a probing experiment seems debatable.
>
> **Q6**
> Thank you for the clarification!
>
> **Q7**
> Thank you for these additional results! I now understand that the Taylor expansion formula is used for uncertainty estimation. I was misled by lines 306–307, which seem to indicate that once adapted, the Taylor expansion is no longer used.
>
> These new results indeed demonstrate that the Taylor expansion provides confidence intervals correlated with prediction uncertainty. While this is an interesting result, it is arguably not as strong as _uncertainty quantification_, as claimed in the main paper.
>
> **W4**
> Neural ODE is indeed very common and easy to use, but its complexity depends on the solving algorithm employed in the backend. In your paper, you mostly used dopri5, which performs six forward calls to the dynamics between each time step. Coupled with the Taylor expansion, this limits your method to relatively simple dynamical systems, as you mention in A.6.
>
> Thank you for clarifying the origin of the spikes! It might have been more helpful to explain this directly in the appendix, rather than replacing it with a smoothed figure.
>
> **W5**
> I am not sure this fully addresses my concerns. If I understand your reply correctly, the standard alternating minimization works well for both t1 and t2 but does not achieve state-of-the-art (SOTA) performance. Hence, you introduced proximal alternating minimization for t2. However, why not apply the same algorithm to t1? I would expect t1 to also benefit from the proximal method.

---

> > ### Author Response · Authors · 2024-11-26
> >
> > **W4** We agree that the Dopir5 solver can be complex, but we are fortunate that Neural ODEs have matured enough that many libraries efficiently abstract away the complexities of implementing differentiable solvers for differential equations [1,2,3]. In addition to those, our work also employs custom simple solvers, notably the fixed time stepper RK4 on Lotka-Volterra, or even Euler on Navier-Stokes (as was done by [4]).
> >
> > Concerning Taylor expansion, we note that our hope with higher-order expansions is to allow efficient modelling of inherently nonlinear problems. We can thus model both non-linear and linear problems in a powerful way. Importantly, we note that previous work presented at this conference outlined the prevalence of __linear__ problems in nature (using similar Taylor expansions) [5]. This emphasizes the broad applicability of our method.
> >
> > [1] https://github.com/rtqichen/torchdiffeq
> >
> > [2] https://github.com/DiffEqML/torchdyn
> >
> > [3] https://github.com/patrick-kidger/diffrax
> >
> > [4] Kirchmeyer et al., Generalizing to New Physical Systems via Context-Informed Dynamics Model, ICML, 2022.
> >
> > [5] Blanke et al. Interpretable meta-learning of physical systems. ICLR, 2024.
> >
> >
> > ---
> >
> > **W5** The reviewer indeed understands correctly. NCF-t1 can benefit from the relatively harder proximal minimization. However, we opted to only present two variants in Table 1 to avoid complexifying our paper any further. We appreciate the simplicity of __ordinary__ minimization for NCF-t1, and we believe most readers of our paper will stop at NCF-t1 (for its simplicity, but also for its proven benefit of interpretability). More interested readers are free to pick and combine the various components of our method, especially since the codebase we provide allows the application of __proximal__ minimization to NCf-t1 with __a single line of code__. Finally, we note that our naming convention is not uncommon, and is used similarly for CoDA-l1 and CoDA-l2 [1].
> >
> > [1] Kirchmeyer et al., Generalizing to New Physical Systems via Context-Informed Dynamics Model, ICML, 2022.
> >
> > ---
> >
> > ### About your overall rebuttal update
> >
> > Thank you for summarizing the rebuttal update. It made addressing your additional concerns easier.
> >
> > Concerning the overall restructuring the reviewer mentioned, we believe this can be easily achieved without significant effort. In line with comments from Reviewer wQzk, we believe replacing section 3.3 with our summarized Interpretability or Uncertainty results should strengthen our work while preserving its flow and message.
> >
> > So far, this rebuttal discussion has served its valuable purpose in that it allowed us to improve our presentation. The changes brought to the main text have been minimal, and we hope to keep it that way. We thank you once more for your invaluable contribution in making all this possible.

---

> ### Author Response · Authors · 2024-11-26
>
> We thank you for acknowledging our rebuttal efforts and recognizing the improved clarity of our work. Most importantly, we thank you for the added time you spared to provide additional feedback. We reply to some of them below.
>
> ---
>
> **Q2**
> Your understanding indeed corresponds to what we wished to convey. We agree that our explanation only based on metrics and datasets looked at the symptoms and not the causes of this behavior. We should add that the various training regularization mechanisms (L1 loss, L2 weight decay, Taylor expansion which implicitly smooths higher-order derivatives, etc.) might also play a role.
>
> Compounded with these regularizations, the fact that the context fine-tuning step is performed on an environment-by-environment basis influences this behavior (see sequential __Algorithm 2__). Indeed, when performed in a bulk all at once (__Algorithm 4__) with Taylor expansion, the OoD performance became much lower. This is what motivated our cautious comments around __line 979__ to encourage users to disable regularization during adaptation.
>
> However, we must emphasize that we did not test this "bulk adaptation" hypothesis with the competing methods that report the same better OoD performance on the same datasets [1,2].
>
> We thank the reviewer for insightful remarks on this question, as the hypothesis we formulated for ours and other methods should benefit from a full exploration in future work, perhaps a full paper investigating parametric PDE methods across the board.
>
> [1] Kirchmeyer et al., Generalizing to New Physical Systems via Context-Informed Dynamics Model, ICML, 2022.
>
> [2] Koupaï et al. GEPS: Boosting Generalization in Parametric PDE Neural Solvers through Adaptive Conditioning, NeurIPS, 2024
>
> ---
>
> **Q4**
> We are happy our explanation helped clarify our method. In __Figure 15__, we agree that the number of data points is small: 9 InD embeddings and 4 OoD embeddings. We note however, that this is a fundamental limitation of the dataset we inherited from [1], which is aligned with the "data scarcity" constraint you referred to. That said, another visualization on the SP problem is displayed in __Figure 13__. We designed its dataset from scratch, and it makes a similar point about context proximity (it contains more data points: 25 InD embeddings, and 2 OoD embeddings).
>
> Concerning the intra-cluster performance, we've investigated this issues on the SM problem, where we show much better performance in the fixed equilibrium (E) with environments $e_3$ and $e_4$ (__Figure 4c__). Furthermore, our new ablation study with __Figure 23 (left)__ should provide insights into how soon in the training these clusters are formed, depending on the pooling strategy. Finally, the contents of clusters and their embeddings can be understandably glanced through the lens of __Proposition 2__, which directly relates these contexts to the underlying physical parameters.
>
> [1] Kirchmeyer et al., Generalizing to New Physical Systems via Context-Informed Dynamics Model, ICML, 2022.
>
> ---
>
> **Q5** What prevents gradient descent from directly identifying $\tilde \xi$ ?
>
> We apologize for not correctly grasping your original question; and we thank you for acknowledging the pertinence of our linear probing experiments. To answer the reviewer's question, we don't see why gradient descent couldn't attempt to directly identify $\tilde \xi$.
>
> However, more than performing a simple non-linear transformation, our grand hope is to lift $\xi$ and $x$ into representational spaces that make it easier for them to interact (like we point out in __line 209__ of the PDF). We note that __lifting__ is a popular approach when solving parametric PDEs, notably used in the now-famous FNO [3]. We've hopefully made this clearer in __line 208__ of the revised PDF.
>
> Our overall approach can also be interpreted as embedding physically-grounded constraints into the model, which ultimately proves economical for parameter count, and performs better.
>
> [3] Li et al. Fourier Neural Operator for Parametric Partial Differential Equations, ICLR, 2021.
>
> ---
>
> **Q7**
> We agree that the wholistic term __uncertainty quantification__ (UQ) covers a much broader range of concepts that we do not cover in this work (data measurements, sources of uncertainty, sensitivity analysis, etc.). A full investigation of UQ would no doubt require a paper of its own. For that reason, you will notice that we've used __uncertainty estimation__ in our revised PDF which is a weaker term, but we believe describes appropriately what we've done. Thank you for your kind words on the corresponding experiment.
>
> ---

---

> ### Author Response · Authors · 2024-12-02
> **Gentle reminder to review our latest responses**
>
> Dear Reviewer 6smX,
>
> Thank you for your thoughtful feedback throughout this review process.
>
> On 26 November 2024, we provided several updates clarifying concerns, questions and remarks you still had. They particularly address the three points that motivate your current overall recommendation, as outlined in your Rebuttal Update.
>
> With the author-reviewer discussion deadline nearly here, could you please review those latest responses ? We deeply appreciate the score improvement from 1 to 5 and hope our revisions, if satisfactory, might lead to an even more favorable adjustment to your score.
>
> We’re also happy to address any further comments or suggestions.
>
> Thanks again!
>
> Authors

---

### Official Review · Reviewer_wQzk · 2024-11-03

**Soundness:** 3
**Presentation:** 2
**Contribution:** 3
**Rating:** 6
**Confidence:** 4

**Summary:**

This work proposes a new meta-learning strategy for learning dynamical systems governed by PDEs. It introduces a new multi-environment framework, where environments are defined by specific PDE coefficients, each describing a specific behavior. To do so, the paper proposes a Taylor expansion of a forecaster network at a context vector $\xi^e$ around other context vectors $\xi^j$. It thus collects information from other environments via a context pool of P indices, modulating the forecaster network and allowing also contexts themselves to be self-modulated. The method has other properties: parallelization, interpretability, uncertainty estimation, ... The method is evaluated both for in-domain and out-domain environments agianst two baselines (CAVIA & CODA) on multiple datasets and show competitive or sota performance.

**Strengths:**

The targeted problem is important. Building neural-ode like solvers able to generalize to changes in the PDE coefficients is important, often referred to solving parametric PDEs.

The method seems novel for learning dynamical systems with changes in pde coefficients. The use of taylor expansion is intuitive and natural. I particularly liked the intuition given line 202-205. Existing context-based methods do not try to leverage information from each context vectors, each describing the environment information. NCF fills this gap.

The method is evaluated on a wide range of PDE problems and is SOTA or competitive when considering a second order talyor expansion.

**Weaknesses:**

Regarding the writing style of the paper:
- I think there is room for improvement. In the introduction, I think the problem of solving parametric PDEs / learning dynamical systems with varying PDE coefficients should be stated more clearly and explains what is an environment in your specific setting. The introduction should 1) clearly defines the problem of building generalizable neural PDE solvers, 2) what are the different directions taken to do so [1, 2, 3] and 3) explain how your work fits into these different directions and advances the field.
- There are especially 2 paragraphs, where neural ODEs and physics-based (hybrid) approaches are introduced, that are not necessary or too much detailed in my opinion.

The authors state that the method can be interpretable, provides uncertainty quantification and is parallelizable. These are important properties that lack for instance for CoDA, as you mentioned. I think that the paper should have exploit these properties in more depth, provide more ablation studies to show that NCF can exploit these properties such as:
- a detailed analysis showing how the learned context vectors relate to physical parameters, demonstrating interpretability
- benchmarks showing how parallelization impacts training time as the number of environments increases

[1] Subramanian et al.,  Towards Foundation Models for Scientific Machine Learning: Characterizing Scaling and Transfer Behavior, NeurIPS, 2023.

[2] Takamoto et al., Learning Neural PDE Solvers with Parameter-Guided Channel Attention, ICML, 2023.

[3] Kirchmeyer et al., Generalizing to New Physical Systems via Context-Informed Dynamics Model, ICML, 2022.

**Questions:**

You introduce different context pool strategy (RA, NF, SF). What are the differences in terms of performance for your method? Some ablations should have been done to compare the different methods, e.g.:
- Comparing the performance (e.g. MSE, adaptation time) of RA, NF, and SF for different datasets.
- Analyzing how the choice of strategy impacts the learned context representations.
Then, propose guidelines for choosing the appropriate strategy.

You mentioned LEADS, a multi-task learning problem. It would have been nice to add this baseline to the different generalization experiments, especially for the new datasets that are not present in [3].

Can you provide training time details for the different meta-learning frameworks?

---

> ### Author Response · Authors · 2024-11-22
>
> We thank the reviewer for taking the time to review our work and for finding important the problem we tackle. We are pleased and honoured to contribute to this important field of solving parametric PDEs. We are happy you found our approach novel and intuitive. We hope the rest of the community will do the same, and will appreciate our SOTA results as did the reviewer.
>
>
> We have addressed the weaknesses and questions in the order the reviewer presented them below.
>
> ---
>
> ### W1. Improvements to our introduction
> We thank the reviewer for this feedback, and for suggesting __specific__ changes that could be added such as the gap our method fills, the references, etc. As a result, we have made several changes to the writing, highlighted in red in the revised PDF. We clearly indicate what an environment is in our case at line 50, whose paragraph (the second in the introduction) illustrates the problem of building generalizable solvers. This lays out the problem with data (P1). The small third paragraph introduces SciML and hybrid approaches that use physics, whose absence raises questions about unobserved parameters (P2). The third paragraph spells out these two problems and why it is important to solve them.
>
> We agree that the paragraph about Neural ODEs in the original PDE wasn't very useful. To that end, it was essentially rewritten (now the fourth paragraph in the revised PDE) to highlight the field of parametric PDE solving, existing methods, and most importantly, __how our methods fits in these__. Our extensive Related Work section was used to formally set up the problem (notations, definitions, equations, etc.) while fleshing out existing Neural ODE-based meta-learners. Finally, we made minor paragraph reordering changes to improve the flow.
>
> ---
>
> ### W2. Capitalising on the unique properties of our method
> Again, we are grateful, and we thank the reviewer for acknowledging the benefits of interpretability, uncertainty estimation, and patballisability offered by our method, but absent in competing approaches.
> - __Concerning interpretability__, we provided an analysis of NCF-t1 in the form of __Proposition 2__ (line 823) which theoretically demonstrates how the physical parameters relate to the learned contexts. We provided a detailed proof which agrees with the subsequent validation experiment. Additionally, we show that our interpretability is robust to noise in the adaptation trajectory (__Figure 7__ with analysis from line 900).
> - __Concerning uncertainty estimation__, we designed and conducted a complete experimental analysis in __Appendix C.3__ (a completely new section), and provided valuable conclusions based on several quantitative metrics we defined.
> - __Concerning parallelization__, we reported a benchmark showing how well our method scales as the number of training environments increases (__Figure 12__, with analysis from line 1394). We see that our training time per epoch is barely impacted when the number of training environments is scaled by factors up to 13. This analysis complements other benchmarks on the scalability with the number of environments scattered throughout the appendix. For that reason, the __Appendix A.6__ coalesces those figures and tables and provides a complete picture.
>
> ---
>
> ### Q1. Analyzing different context pooling strategy
>
> We have addressed this questions primarily by adding an ablation study in __Appendix D.3__. The associated __Table 10__ and __Figure 23__ indicate that on the LV problem on which we have deeper and more interpretable understanding of the contexts and their impact, different strategies offer different benefits in terms of train times and/or MSEs. We realize that we hadn't clearly indicated in the original PDF that the comments in section 3.3 were based on our intuitive understanding of the method. We have adjusted our wording in the revised PDF, and we only suggest the well-balanced NF when there is evidence to support that (i.e., based on our ablation study in D.3). Our wording also makes it clear that this strategy is indeed an additional __tunable__ hyperparameter (lines 315 and 519), thus constituting a limitation of our method, which is among the problems that will be explored deeper in future work.
>
> ---

---

> ### Author Response · Authors · 2024-11-22
>
> ### Q2. Adding the LEADS baseline
> We agree with the reviewer that it would have been nice to add to our meta-learning comparison the multi-task-learning LEADS baseline for the generalization problems. And in response, we have added the baseline to the two new datasets highlighted as being the most important by the reviewer. We used the reference implementation from [1] and its default hyperparameters. We made sure to increase the capacity of its _left_ model to roughly 50k parameters on SM, and to 308k for BT. The results are presented in the table below. We see that LEADS' performance lies between that of CAVIA and CoDA on the SM problem, but is slightly better (second-best) on the harder BT PDE problem. The modified LEADS code is attached as part of our revised PDF submission.
>
> |          | SM                  | BT                  |
> | -------- | ------------------- | ------------------- |
> | # Params | 50402               | 118325              |
> | InD      | 5.50e-01±3.71e-02   | 1.031e+00±7.024e-01 |
> | OoD      | 3.523e-02±2.705e-03 | 9.178e-01±2.178e-01 |
>
>
> [1] Yin et al. LEADS: Learning dynamical systems that generalize across environments, NeurIPS, 2021.
>
> ---
>
> ### Q3. Training times for different meta-learning frameworks
>
> The meta-training times are reported below in __minutes__ (rounded to the nearest minute after which the validation loss stopped decreasing and the best model was considered). These training time can be complemented with those provided for NCF as it compares to non-meta-learning approaches (see __Table 8__ of the revised PDF).
>
> |        | LV  | GO  | SM  | BT  | GS  | NS  |
> | ------ | --- | --- | --- | --- | --- | --- |
> | CAVIA  | 44  | 50  | 40  | 22  | 136 | 45  |
> | CoDA   | 57  | 98  | 60  | 30  | 202 | 42  |
> | NCF-t1 | 59  | 104 | 45  | 56  | 78  | 22  |
> | NCF-t2 | 58  | 171 | 80  | 115 | 184 | 23  |
>
> ---
>
> We thank you once more, and we hope these clarify your questions. We look forward to a fruitful discussion in case we left anything unanswered.

---

> > ### Comment · Reviewer_wQzk · 2024-11-25
> >
> > I would like to thanks the authors for improving the quality of the paper and the added experiments.
> >
> > I particularly appreciated the extended results added in appendix and the new theorotical results to identify the physical parameters. Despite all results are not particularly impressive (e.g. concerning parallelization), it still improves the strength of the paper overall in my opinion.
> >
> > Concerning the variation of the pooling strategy, it looks like there is not a strategy that performs best between RA and NF, at least for the LV problem. Therefore, I am not sure of the relevance of this section, at least in the main work. There are some sections in appendix (e.g. interpretability and uncertainty estimation) that could be included in the main paper, instead of the section 3.3.
> >
> > It is nice that you also added the LEADS baseline.
> >
> > Concerning the presentation of the paper, I still have some small concerns, notably how environments are introduced. It should be explicitly said that environments here correspond to changes in the parameters $c$ of the differential equation, and maybe diretly linked to the work on parametric PDEs (I think that defining environments through an example is a bit superficial).
> >
> > Overall, my concerns remain small and can easily be changed for the final version. Therefore, I upgraded my score (confidence, rating and soundness).

---

> > > ### Author Response · Authors · 2024-11-26
> > >
> > > Dear reviewer, we thank you again for your time reviewing our paper and for acknowledging our rebuttal.
> > >
> > > We agree that apart from the _speed_ of convergence, the best between RA and NF is not easy to pick. We agree that other sections might benefit from more space in the main paper, and we will take great care to address this in the final version. A swap of section 3.3 with more Interpretability or Uncertainty material from the Appendix should indeed leave the main text minimally impacted, and keep the page count below 10. As suggested, we will also find clever ways to introduce environments.
> > >
> > > Thank you very much for suggesting these changes.

---

### Official Review · Reviewer_oqLH · 2024-11-04

**Soundness:** 3
**Presentation:** 4
**Contribution:** 3
**Rating:** 8
**Confidence:** 4

**Summary:**

The paper introduces Neural Context Flows, a method for meta learning. The main contributions of the work are focusing on how to combine context vectors in a way that allows OoD generalization and interpretability.

**Strengths:**

- Writing is very clear and the methodology is well explained. This allows readers to understand the differences between this method and previous ones.
- Interesting use of context vectors through the 3-network model. Ablation studies in supplementary material show the need for such an architecture.
- The combination via context through the Taylor expansion seems to be an interesting and novel application, which I can see being used in other fields outside of ODE and PDE simulations.
- The estimation of uncertainty via different context vectors is very simple yet very clear and useful.

**Weaknesses:**

Manuscript makes reference to sample efficiency of using such adaptive models for new context. However the manuscript does not include any experiments to support such a statement.

**Questions:**

Question:

- With respect to the 3-network model, can you remove the state-network but keep the context network? Does it make any difference compared to the one-network model which performs similar to CAVIA?


Suggestions:

- Include sample efficiency experiments for some example ODEs and PDEs. For example MSE for a model trained from scratch vs one finetunes via a new context vector.
- Include uncertainty as a function of forecast. One would assume that uncertainty increases as the forecast becomes longer. Could you provide such an estimate?

---

> ### Author Response · Authors · 2024-11-22
>
> We thank the reviewer for their examination of our paper, along with the kind words regarding our writing, our methodology, the various benefits it offers, and its potential for application outside ODE and PDE simulation.
>
> Regarding the weaknesses, questions and suggestions raised by the reviewer, we've addressed them all in the revised PDF we've uploaded. We summarize our answers below.
>
> ---
>
> ### Q1. Can we remove the state network, but keep the context network ?
> We've done this with NCF-t1 on the SP and LV problems. On both problems, we directly concatenated the output of the context network $\tilde \xi$ to the state $x$ before feeding into the main network. To keep the comparison fair, we increased the hidden units of the two networks to match the ordinary NCF parameter count of 50k for SP and 308k for LV (as observed in __Table 1__). We report the results in the table below, with NCF* indicating this variant of NCF without a state network (i.e. a two-network architecture). We notice remarkably lower performance on both problem (by almost an order of magnitude for training, and more for adaptation). Given that the system state is typically much lower-dimensional compared to the output of the context network (in this case $d_x=2$ for both, whereas $d_{\tilde \xi}=82$ for SP and $d_{\tilde \xi}=74$ for LV), this significant drop in performance might be explained by the idea that the network relies considerably more on contextual information rather than looking for commonalities in the environments' states. All this further motivates our intuition that contexts and states should be pre-processed into similar spaces before they can interact.
>
> |            | SP                  | LV              |
> | ---------- | ------------------- | --------------- |
> | NCF*-Train | 1.04±0.2            | 4.56e-4±0.7e-4  |
> | NCF*-Adapt | 0.11±0.06           | 5.31e-2±1.89e-3 |
> | NCF-Train  | 0.01± 0.003         | 6.73e-5±0.87e-5 |
> | NCF-Adapt  | 0.0000356± 0.000001 | 7.92e-5±1.04e-5 |
>
> Furthermore, like the reviewer pointed out, deleting both context and state network results in an architecture much similar to the original CAVIA [1]. That said, on the Navier-Stokes problem, we show that our two-networks model performs better than a CAVIA similarly equipped with a context network (cf. __lines 1074 and 1176__ of the revised PDF). (This was our only experiment containing no state network in our original PDF). This highlights the benefits of our Taylor-based self-modulation, which is absent in CAVIA.
>
> Further details regarding our ablation of the 3-networks architecture were added to the appendix D.4 of the revised PDF. We also reran and updated the base SP problem with a bigger step size, and corrected a few typographical errors.
>
> [1] Zintgraf et al. Fast Context Adaptation via Meta-Learning, ICML 2019.
>
> ---
>
> ### Q2. Sample efficiency, training from scratch vs context fine-tuning
> We thank the reviewer for suggesting this addition. We have addressed this by complementing our One-For-All vs NCF comparison with the One-Per-Env paradigm (one model trained from scratch on each environment). The details of the corresponding experiment are presented in __Table 2__ and __Appendix C.3__ starting around line 1337. On the noticeably hard SP problem, they show that OPE is time-consuming and overfits to its 4 InD trajectories, and performs even worse on the one-shot OoD trajectory.
>
> Regarding the __sample efficiency__ specifically, we consider the SP problem (__Figure 11__) and BT problems (__Figure 18__). We find that as the number of trajectories increases, One-Per-Env's performance improves, even outperforming NCF on the SP ODE problem. Importantly, we show that NCF remains the most efficient choice in low data-data regimes.
>
> ---
>
> ### Q3. Uncertainty as a function of forecast time.
> We thank the reviewer for this question, which motivated us to investigate uncertainty estimation further. As a result, we conducted a deeper analysis of uncertainty quantification with NCFs, and the results are presented in __Appendix C.2__. Specifically, __Figure 8__ shows that __uncertainty grows with forecast time__, thus confirming the reviewer's intuition.
>
> ---
>
> Once again, thank you for your positive and valuable feedback. These insights have helped us improve the clarity of our paper.

---

> > ### Author Response · Authors · 2024-11-30
> > **Gentle reminder for rebuttal acknowledgment**
> >
> > Dear Reviewer oqLH,
> >
> > Thanks again for your helpful feedback. We've carefully addressed all your comments and updated our manuscript, particularly around __Appendices C.2__ and __C.3__.
> >
> > Since the author-reviewer discussion deadline is coming up, we’d be grateful if you could review our responses. If satisfactory, we hope they inspire an even more favorable adjustment to your original positive rating.
> >
> > We’re happy to address any other questions or remarks you might have too.
> >
> > Thanks a lot!
> >
> > The Authors

---

### Author Response · Authors · 2024-11-23

Dear reviewers,

We are enormously grateful to all of you for taking the time to review and comment on our work. We are happy you all _generally_ found our paper clearly written, with intuitive explanation that helped understand the subject. We are happy the novelty of our approach shone through, especially since our theoretical insights were validated through extensive experimentation and produced SoTA results, all while exhibiting exceptional benefits such as _interpretability_, _massive parallelisability_, and _uncertainty estimation_.

Hoping to have a great discussion, we have taken great care to address all of your concerns. Although our main text remains largely unchanged, additional experiments resulted in several additional pages in the appendix. Figure, Table, and Section numbers were all affected. We summarize the major changes as follows, using the numbering in our revised PDF (in which new or modified material is highlighted in red):
- __Appendix A.2__ provides theoretical demonstration of the interpretability of our method
- __Figure 7__ shows robustness of our method to noise in the adaptation trajectory
- __Appendix C.2__ provides a broad account of uncertainty estimation with our method, with several quantitative metrics used to provide meaningful uncertainties.
- __Table 8__ and __Figure 10__ compare our method to non-meta-learning baselines, and highlight the major benefits of meta-learning.
- __Figures 11 and 18__ highlight sample efficiency with the number of trajectories per environments.
- __Figures 12__ highlights excellent scaling with the number of training environments.
- __Appendix D.3__ investigates the effect of changing our context-pool filling strategy.
- Finally, since we added a new proposition, we found it important to rename the existing Proposition 3.1 to __Proposition 1__, leaving room for __Proposition 2__ (the new one). Similarly, Theorem 3.1 became __Theorem 1__.

We thank you once more for suggesting them. We've stressed the value of these changes in our individual responses to you all. We clearly see how they emphasize our method's strengths, and we hope they clarify your concerns.

Thank you,

The Authors

---

### Meta-Review · Area_Chair_WANd · 2024-12-18

**Metareview:**

This paper proposes a novel meta-learning strategy for modeling dynamical systems across varying environment parameters. The core idea is to introduce environment-specific latent context vectors and expand the vector field using a Taylor series about these context vectors. This approach facilitates information sharing across environments, enhancing data efficiency and improving adaptation to new environments. The method is evaluated for both in-domain and out-of-domain environments against several state-of-the-art meta-learning baselines on multiple datasets, demonstrating competitive performance.

All reviewers agree that the targeted problem is important, and the use of Taylor expansion is intuitive and well-supported. Given the promising applications in meta-learning and the compelling results, I recommend this paper for publication.

**Additional Comments On Reviewer Discussion:**

The reviewers requested additional numerical results for clarification, including comparisons with non-meta-learning methods, robustness to noise, ablation studies on the number of trajectories per environment, the number of training environments, and the context pooling strategy. The authors made significant efforts to provide more convincing empirical results and some theoretical justification. Most concerns were addressed, and the manuscript has been improved for greater clarity.

---

### Decision · Program_Chairs · 2025-01-22

Accept (Poster)